# Fixed-Point RNNs:
# Interpolating from Diagonal to Dense

**Sajad Movahedi**[* 1,2], **Felix Sarnthein**[* 1,2], **Nicola Muça Cirone**[3], **Antonio Orvieto**[1,2]

[1]ELLIS Institute Tuebingen, [2]Max Planck Institute for Intelligent Systems,
[3]Department of Mathematics, Imperial College London
{sajad.movahedi, felix.sarnthein}@tue.ellis.eu

## Abstract

Linear recurrent neural networks (RNNs) and state-space models (SSMs) such as Mamba have become promising alternatives to softmax-attention as sequence mixing layers in Transformer architectures. Current models, however, do not exhibit the full state-tracking expressivity of RNNs because they rely on channel-wise (i.e. diagonal) sequence mixing. In this paper, we investigate parameterizations of a large class of dense linear RNNs as fixed-points of parallelizable diagonal linear RNNs. The resulting models can naturally trade expressivity for efficiency at a fixed number of parameters and achieve state-of-the-art results on the state-tracking benchmarks $A_5$ and $S_5$, while matching performance on copying and other tasks.

## 1 Introduction

State-space models (SSMs) and other new efficient recurrent token mixers are becoming a popular alternative to softmax attention in language modeling (Gu & Dao, 2024) as well as in other applications such as vision (Liu et al., 2024) and DNA processing (Nguyen et al., 2024). Inspired by linear input-controlled filtering, these models can be expressed as carefully parametrized linear recurrent neural networks (RNNs) with input-dependent, diagonal state transition:

$$\mathbf{h}_t = \mathrm{diag}(\mathbf{a}_t)\mathbf{h}_{t-1} + \mathbf{B}_t\mathbf{x}_t \qquad (1)$$

Compared to classical RNNs such as LSTMs (Hochreiter & Schmidhuber, 1997), in Eq. (1) the relation between the previous hidden state $\mathbf{h}_{t-1}$ and the current $\mathbf{h}_t$ is linear and its coefficient $\mathbf{a}_t$ does not depend on the hidden states. These choices allow SSMs such as Mamba (Gu & Dao, 2024) to be computed through efficient parallel methods during training. Furthermore, they are easier to optimize than classical RNNs, thanks to stable and efficient reparametrizations available for diagonal transitions (Orvieto et al., 2023; Zucchet & Orvieto, 2024) – techniques that are significantly more difficult to apply effectively in the classical setting (Arjovsky et al., 2016; Helfrich et al., 2018). At test time, they are faster than classical Transformers on long sequences due to their recurrent nature.

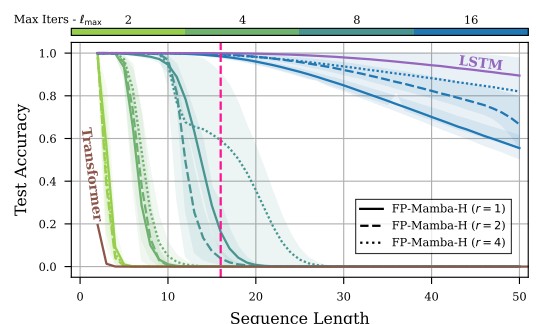

Figure 1: *Sequence length generalization at training length 16 (pink) for state-tracking on $A_5$, with Transformer (brown) and LSTM (purple) as lower/upper bounds. Our Fixed-Point RNN (FP-Mamba-H) is trained at different maximum number of fixed-point iterations $\ell_{max}$: between 2 (green) and 16 (blue).* **Increasing the number of fixed-point iterations allows the linear RNN to interpolate from diagonal to dense in a few iterations.**

---

[*]Equal contribution.

39th Conference on Neural Information Processing Systems (NeurIPS 2025).

Though modern linear RNNs have shown promise in practice, recent theoretical studies suggest that using dense, input-dependent transition matrices (i.e. replacing diag($\mathbf{a}_t$) with a dense $\mathbf{A}_t$) could present an opportunity to improve expressivity and unlock performance on challenging tasks. In particular, Cirone et al. (2024b) prove that dense selective SSMs are endowed with the theoretical expressivity of classical non-linear RNNs such as LSTMs. As shown by Merrill et al. (2024) and Sarrof et al. (2024), such gained expressivity proves to be particularly useful in state-tracking applications where models are expected to maintain and extrapolate a complex state of the world. Since state-tracking is naturally expressed by non-linear RNNs but provably unavailable to channel-wise sequence mixers such as SSMs or Transformers, Merrill & Sabharwal (2023) speculate on a *fundamental tradeoff between parallelism and expressivity*. This discussion sparked interest in non-diagonal recurrences and parallelizable architectures capable of state-tracking (Grazzi et al., 2024; Terzic et al., 2025; Schöne et al., 2025; Peng et al., 2025; Siems et al., 2025).

When designing new architectures involving dense selective yet *linear* state transitions of the form $\mathbf{h}_t = \mathbf{A}_t \mathbf{h}_{t-1} + \mathbf{B}_t \mathbf{x}_t$, two fundamental concerns arise:

1. What should the parametric form for $\mathbf{A}_t$, as a function of the input be? How can we guarantee this parametrization induces a stable recurrence, like in standard[2] SSMs?

2. How does a parametrization balance between expressivity and parallelism? Which assumptions on the structure of $\mathbf{A}_t$ enable efficient computation, and how do they interact with expressivity?

Perhaps the first approach tackling the above questions was DeltaNet (Schlag et al., 2021a; Yang et al., 2024b) with a block-diagonal and orthogonal therefore, stable state transition structure, where each block is parametrized by a Householder matrix. The parallelizable algorithm, was then extended to include negative eigenvalues (Grazzi et al., 2024), gates (Yang et al., 2025), and most recently products of Householders (Siems et al., 2025). Such choices, leading to increased expressivity as exemplified by their state-tracking and length generalization capabilities, are motivated mainly by hardware considerations: Householder-based mixing can be implemented efficiently on GPUs as linear attention via WY-representations and the UT transform (Yang et al., 2024b).

While the works above offer exciting practical strategies for boosting capabilities at a relatively low additional computational cost, they fall short in exploring the sea of intriguing options for dense transitions and hence, in thoroughly answering questions (1) and (2) above.

Unfortunately, this is not an easy task: although linear recurrences are theoretically parallelizable across sequence length (Martin & Cundy, 2018), parallelizing dense RNNs efficiently is not trivial due to increased memory I/O. These thoughts inspired us to change our viewpoint: instead of designing an algorithm which adds a fraction of non-diagonal processing to a model, here, we look for a strategy to navigate the *parallelism tradeoff* towards a truly dense object.

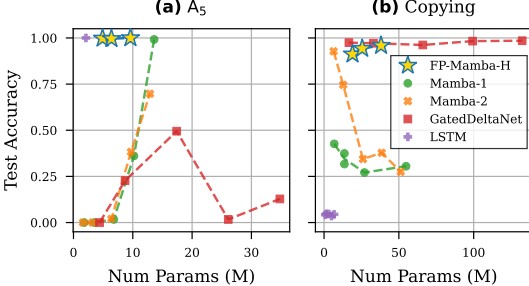

Figure 2: *(a) State-tracking on $A_5$ at sequence length 16, and (b) character accuracy of copying at $2\times$ sequence length generalization, trained on lengths $\in [5, 50]$. Our single layer FP-Mamba-H with mixer reflections $r \in \{1, 2, 4\}$ is compared to baselines of increasing depth $\in \{1, 2, 4, 6, 8\}$.* **FP-Mamba-H is the only model capable of solving both the state-tracking and the copy task.**

Motivated by the idea of designing a parallelizable general-purpose method to implement new dense RNN variations, in this paper we devise a new adaptive computation strategy which allows to interpolate between fast recurrent diagonal RNNs and dense recurrences with arbitrary preselected structure. Instead of parametrizing the dense RNN layer as an *explicit* function $\mathbf{h} = F_{\boldsymbol{\theta}}(\mathbf{x})$, we build on the literature of equilibrium/implicit models (Bai et al., 2019; Ghaoui et al., 2021) to parametrize it *implicitly* as a solution $\mathbf{h}^*$ to a fixed-point equation $\mathbf{h} = f_{\boldsymbol{\theta}}(\mathbf{x}, \mathbf{h})$ involving only a diagonal RNN. As described in Fig. 3a, we solve for $\mathbf{h}^*$ using a fixed-point iteration of diagonal RNN evaluations $f_{\boldsymbol{\theta}}$.

---

[2]Standard SSMs are diagonal and operate in polar coordinates, parametrizing directly the gap between eigenvalues and the stability threshold (Orvieto et al., 2023). This technique allows to increasing granularity near the identity, and to effectively normalize the forward pass (cf. $\sqrt{1 - |\gamma|^2}$ term in Griffin (De et al., 2024)).

A fundamental question some readers might rightfully ask, is the following: "*what is the advantage of iterating a single layer in depth compared to depth-stacking multiple SSM, e.g. Mamba layers?*" We claim one advantage comes from having access to the limiting dense object. As showcased by Fig. 2, this allows to adaptively provide the required expressivity for a fixed set of parameters without any a priori choice on the network size.

**Summary.** In this work, we propose a recipe to design a general class of dense linear RNNs as fixed points of corresponding diagonal linear RNNs. Our contributions are:

1. We develop the framework of Fixed-Point RNNs to adaptively trade parallelism for expressivity using the number of fixed-point iterations (Fig. 1).
2. We achieve a stable parametrization of a dense RNN via a carefully designed diagonal RNN.
3. The framework allows for easy integration of both non-linear hidden state dependence and linear attention based matrix-valued formulations. This way, our FP-Mamba unites previously isolated capabilities of recurrent computation and memory (Fig. 2).

## 2 Background

Since their introduction (Rumelhart et al., 1986; Elman, 1990), RNNs have significantly contributed to the evolution of machine learning methods for sequential data (Hochreiter & Schmidhuber, 1997; Jaeger, 2001). But despite their theoretical promise of Turing-completeness (Siegelmann & Sontag, 1992), recurrent models fell out of fashion due to two significant challenges: they are inherently sequential, and notoriously difficult to train (Hochreiter et al., 2001; Pascanu et al., 2013). The recent advancements of linear RNNs (Gu & Dao, 2024) suggest a way forward to combine the scalability of Transformers (Vaswani et al., 2017) with the expressivity of classical RNNs (Cirone et al., 2024b). The key challenge here is the stable and efficient parametrization of a linear RNN layer with a time-varying recurrent transition matrix. In this paper we are exploring first steps towards this goal.

**Dense Selective RNN.** Traditionally, RNNs are parametrized as either time-invariant, non-linear, or element-wise system. To the best of our knowledge, a time-variant, dense, and linear RNN parametrization has been of mild interest at best. To understand why, consider the general form

$$F_{\boldsymbol{\theta}} : \mathbf{x} \mapsto \mathbf{h}, \qquad\qquad \mathbf{h}_t = \mathbf{A}_t \mathbf{h}_{t-1} + \mathbf{B}_t \mathbf{x}_t, \qquad\qquad (2)$$

where $\mathbf{A}_t \in \mathbb{R}^{d \times d}$ corresponds to the time-varying state transition matrix, $\mathbf{B}_t \in \mathbb{R}^{d \times d}$ is the input transformation matrix, $\mathbf{h}_t \in \mathbb{R}^d$ denotes the hidden state, and $\mathbf{x}_t \in \mathbb{R}^d$ is the input for $t < T$ steps. For a given sequence of $\mathbf{A}_t$, the complexity of a forward pass is $O(Td^2)$ in memory and $O(T)$ sequential steps. Although such a linear RNN could also be computed in $O(\log T)$ sequential steps using a parallel scan algorithm (Martin & Cundy, 2018), this would require materializing matrix-matrix multiplications at cost $O(d^3)$. An issue in both scenarios, however, is the parametrization of $\mathbf{A}_t$ as time-varying, i.e. input- or even hidden state-dependent matrices. In general, this requires a map $\mathcal{M} : d \mapsto d^2$ with potentially $d \times d^2$ parameters, and $O(Td^3)$ time complexity. While structured dense matrix representations for $\mathbf{A}_t$ could potentially present a remedy, they come with additional challenges: **(1)** In order to guarantee expressivity, the $\mathbf{A}_t$ cannot be co-diagonalizable such as for example Toeplitz matrices (Cirone et al., 2024b). **(2)** In order to guarantee stability of the dynamical system, the spectral radius $\rho(\mathbf{A}_t)$ needs to be less than, but still close to 1 for long-range interactions (Orvieto et al., 2023). **(3)** The matrix structure needs to be closed under multiplications to enable parallel scans without having to materialize dense representations at $O(Td^2)$ memory cost.

**Related Works.** Improving the trainability of classical non-linear RNNs has a long history. For example, Arjovsky et al. (2016) and Helfrich et al. (2018) investigate parameterizations to stabilize their spectral radius with structured matrix representations, while Lim et al. (2024) and Gonzalez et al. (2024) propose iterative methods to parallelize their computation. In this work, however, we focus on stabilizing and parallelizing a time-variant, dense, linear RNN. Improving the limited expressivity of existing diagonal linear RNNs is the focus of a few recent works, e.g. by Grazzi et al. (2024) and Siems et al. (2025). In contrast, we investigate a wide class of structured parameterizations for dense RNNs where the additional cost is adaptively chosen depending on the task. In concurrent work, Schöne et al. (2025) propose an iterative method similar to ours, but as opposed to our carefully designed implicit dense RNN layer, they focus on scaling implicit causal models of existing multi-layer architectures on language. For a more extensive literature review, we refer the reader to App. A.

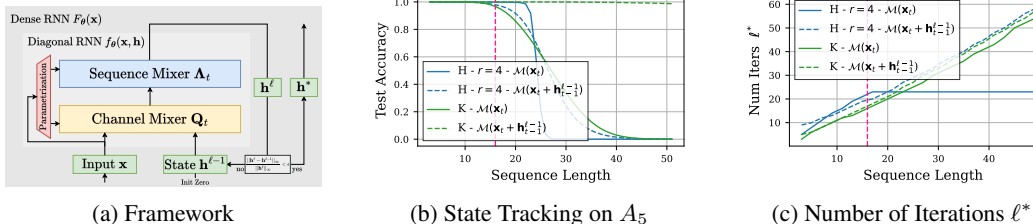

| (a) Framework | (b) State Tracking on $A_5$ | (c) Number of Iterations $\ell^*$ |

Figure 3: **(a)** An overview of the proposed Fixed-Point RNN framework in Sec. 3. A diagonal RNN $f_{\boldsymbol{\theta}}$ consisting of a sequence mixer $\boldsymbol{\Lambda}_t$ and a channel mixer $\mathbf{Q}_t$ is iterated until convergence towards the hidden states of an implicitly dense RNN $F_{\boldsymbol{\theta}}$. **(b)** FP-RNN variants with channel mixer introduced in Sec. 3.3 and 3.4 solve the state-tracking task $A_5$ up to various sequence lengths. **(c)** FP-RNNs adapt their computation time to the difficulty of the task by varying the number of fixed-point iterations $\ell^*$ .

## 3    Fixed-Points as an RNN Layer

In this section, we introduce an implicit parameterization for a family of dense RNNs $F_{\boldsymbol{\theta}}(\mathbf{x})$ which describes its output by a solution $\mathbf{h}^* \in \mathbb{R}^{T \times d}$ to the fixed-point equation $\mathbf{h} = f_{\boldsymbol{\theta}}(\mathbf{x}, \mathbf{h})$ (Sec. 3.1). Then, we discuss how to find the solution $\mathbf{h}^*$ using fixed-point iterations (Sec. 3.2) and the algorithmic implications (Sec. 3.4) of the FP-RNN framework in light of the challenges outlined in Sec. 2. Finally, we briefly touch on how to train an implicitly dense model $F_{\boldsymbol{\theta}}(\mathbf{x})$ with gradient descent (Sec. 3.5).

### 3.1    From Explicit to Implicit Parameterization

We start by designing a diagonal RNN $f_{\boldsymbol{\theta}}(\mathbf{x}, \mathbf{h})$ such that the solution $\mathbf{h}^*$ to its fixed-point equation $\mathbf{h} = f_{\boldsymbol{\theta}}(\mathbf{x}, \mathbf{h})$ implicitly represents a dense RNN $\mathbf{h}^* = F_{\boldsymbol{\theta}}(\mathbf{x})$. Consider the factorized parametrization of $\mathbf{A}_t$ similar to the one introduced by Helfrich et al. (2018) for non-linear and time-invariant RNN:

$$F_{\boldsymbol{\theta}} : \mathbf{x} \mapsto \mathbf{h}^*, \qquad \mathbf{h}_t^* = \mathbf{Q}_t^{-1} \boldsymbol{\Lambda}_t \mathbf{h}_{t-1}^* + \mathbf{B}_t \mathbf{x}_t. \qquad (3)$$

Separating $\mathbf{A}_t$ into a diagonal matrix $\boldsymbol{\Lambda}_t \in \mathbb{R}^{d \times d}$ and a non-diagonal invertible mixing matrix $\mathbf{Q}_t \in \mathbb{R}^{d \times d}$ allows to describe $\mathbf{h}^*$ by only a diagonal transition $\boldsymbol{\Lambda}_t$ by reformulating Eq. 3 to

$$\mathbf{h}_t^* = \boldsymbol{\Lambda}_t \mathbf{h}_{t-1}^* + \mathbf{Q}_t \mathbf{B}_t \mathbf{x}_t + (\mathbf{I} - \mathbf{Q}_t) \mathbf{h}_t^*. \qquad (4)$$

This means that the states $\mathbf{h}^* = F_{\boldsymbol{\theta}}(\mathbf{x})$ of the *dense linear RNN* can be implicitly described by the fixed-point $\mathbf{h}^* = f_{\boldsymbol{\theta}}(\mathbf{x}, \mathbf{h}^*)$ of a corresponding *diagonal linear RNN* of the following form:

$$f_{\boldsymbol{\theta}} : (\mathbf{x}, \mathbf{h}) \mapsto \mathbf{h}', \qquad \mathbf{h}_t' = \boldsymbol{\Lambda}_t \mathbf{h}_{t-1}' + \mathbf{Q}_t \mathbf{B}_t \mathbf{x}_t + (\mathbf{I} - \mathbf{Q}_t) \mathbf{h}_t. \qquad (5)$$

In other words, if we could find the fixed-point $\mathbf{h}^* = f_{\boldsymbol{\theta}}(\mathbf{x}, \mathbf{h}^*) \in \mathbb{R}^{T \times d}$ for the diagonal RNN defined in Eq. 5, then $\mathbf{h}^*$ would describe the states of a corresponding dense RNN $\mathbf{h}^* = F_{\boldsymbol{\theta}}(\mathbf{x})$. Motivated by this insight, in Sec. 3.2 we carefully parametrize the diagonal RNN $f_{\boldsymbol{\theta}}(\mathbf{x}, \mathbf{h})$ and its *channel mixer* $\mathbf{Q}_t$ such that a computable fixed-point exists.

### 3.2    The Fixed-Point Iteration

Solving fixed-point equations such as $\mathbf{h} = f_{\boldsymbol{\theta}}(\mathbf{x}, \mathbf{h})$, is perhaps one of the most well-studied problems in mathematics (Granas et al., 2003). In the context of deep learning, the literature on Neural ODEs (Chen et al., 2018) and Deep Equilibrium Models (Bai et al., 2019; Ghaoui et al., 2021) investigates fixed-point methods for implicit parametrizations of neural networks. A straightforward, yet effective method computes the forward pass by simply rolling out the fixed-point iteration. In the context of solving $\mathbf{h}^* = f_{\boldsymbol{\theta}}(\mathbf{x}, \mathbf{h}^*)$, this corresponds to introducing an iteration in depth $\mathbf{h}^\ell = f_{\boldsymbol{\theta}}(\mathbf{x}, \mathbf{h}^{\ell-1})$. Denoting $\ell$ as the current iteration in depth (i.e., over the layer dimension), and $t$ as the current iteration in time (i.e., over the sequence dimension), the iteration starts at $\mathbf{h}_t^0 = 0$ and proceeds with

$$\mathbf{h}_t^\ell = \boldsymbol{\Lambda}_t \mathbf{h}_{t-1}^\ell + \mathbf{Q}_t \mathbf{B}_t \mathbf{x}_t + (\mathbf{I} - \mathbf{Q}_t) \mathbf{h}_t^{\ell-1}. \qquad (6)$$

Intuitively, this iteration mixes information with interleaved channel mixing (with $\mathbf{Q}_t$) and sequence mixing (with $\boldsymbol{\Lambda}_t$) until convergence towards the hidden states of an implicit dense RNN $F_{\boldsymbol{\theta}}$ (cf. 3a).

The difficulty with such an iteration in *depth and time* is that the recurrent dynamics could explode without proper stabilization. While the recurrence in time can be stabilized with RNN techniques (Zucchet & Orvieto, 2024) such as an input gate $\mathbf{I} - \mathbf{\Lambda}_t$, the recurrence in depth, however, could still diverge if $f_{\boldsymbol{\theta}}(\mathbf{x}, \mathbf{h})$ does not have an attracting fixed-point (Granas et al., 2003). In order to design a diagonal linear RNN $f_{\boldsymbol{\theta}}(\mathbf{x}, \mathbf{h})$ which is guaranteed to have an attracting fixed-point, we make use of Banach (1922)'s theorem. In our context, the theorem states that $f_{\boldsymbol{\theta}}(\mathbf{x}, \mathbf{h})$ converges to a fixed-point from any initialization $\mathbf{h}^0$ if it has a Lipschitz constant $< 1$ in $\mathbf{h}$. For a fixed-point RNNs with input gate $\mathbf{I} - \mathbf{\Lambda}$, we present the following theorem:

**Theorem 3.1.** *Let $f_{\boldsymbol{\theta}}(\mathbf{x}, \mathbf{h})$ be the diagonal linear RNN with input-independent $\mathbf{\Lambda}$ and $\mathbf{Q}$*

$$f_{\boldsymbol{\theta}} : (\mathbf{x}, \mathbf{h}) \mapsto \mathbf{h}', \qquad \mathbf{h}'_t = \mathbf{\Lambda} \mathbf{h}'_{t-1} + (\mathbf{I} - \mathbf{\Lambda})(\mathbf{Q}\mathbf{B}_t \mathbf{x}_t + (\mathbf{I} - \mathbf{Q})\mathbf{h}_t). \qquad (7)$$

*If $||\mathbf{\Lambda}||_2 < 1$ and $||\mathbf{I} - \mathbf{Q}||_2 < 1$, then $f_{\boldsymbol{\theta}}(\mathbf{x}, \mathbf{h})$ has a Lipschitz constant $< 1$ in $\mathbf{h}$. Proof in App. B.1.*

Intuitively, Thm. 3.1 states two conditions for stable parametrization of an implicitly dense RNN $F_{\boldsymbol{\theta}}$: **(1)** the recurrence in time needs to be coupled with input normalization and contractive (i.e. $||\mathbf{\Lambda}||_2 < 1$). **(2)** The recurrence in depth acting on $\mathbf{h}$, i.e. $(\mathbf{I} - \mathbf{Q}_t)$, needs to be contractive. Together, this guarantees that all sequences $\mathbf{h}^\ell$ up to $\mathbf{h}^*$ throughout the fixed-point iteration do not explode without any explicit assumptions on the spectral radius on $\mathbf{A}$ (Arjovsky et al., 2016).

## 3.3 Parametrization of $\mathbf{Q}_t$ and $\mathbf{\Lambda}_t$

To satisfy the assumptions required for expressivity in (Cirone et al., 2024b), the implicit transition matrix $\mathbf{A}_t$ and therefore $\mathbf{\Lambda}_t$ and $\mathbf{Q}_t$ need to be input-controlled (i.e. selective), which could be realized through a linear mapping of the input, i.e. $\mathbf{Q}_t = \mathcal{M}(\mathbf{x}_t) := \text{reshape}(\mathbf{W_Q}\mathbf{x}_t)$. However, this presents two challenges: how can stability be guaranteed (c.f. Thm. 3.1) and excessive computational cost due to the $O(d^3)$ parameters of $\mathbf{W_Q}$ be avoided? A straight-forward solution lies in structured matrix representations for both the diagonal transition matrix $\mathbf{\Lambda}_t$ and the channel mixer $\mathbf{Q}_t$.

Inspired by Helfrich et al. (2018), we aim for $\mathbf{Q}_t$ to be approximately norm-preserving and $\mathbf{\Lambda}_t$ to control the eigenvalue scale using a parametrization akin to Mamba or Griffin (Gu & Dao, 2024; De et al., 2024) and normalization $(\mathbf{I} - \mathbf{\Lambda}_t)$. For the channel mixers $\mathbf{Q}_t$, we consider the structures:

- **Diagonal Plus Low Rank (DPLR):** $\mathbf{Q}_t = \mathcal{M}(\mathbf{x}_t) := \left(\mathbf{I} - \sum_{i=1}^r \alpha_{it} \cdot \bar{\mathbf{u}}_{it} \bar{\mathbf{u}}_{it}^\top\right)$, for rank $r$.
- **Householder Reflections (H):** $\mathbf{Q}_t = \mathcal{M}(\mathbf{x}_t) := \prod_{i=1}^r \left(\mathbf{I} - \alpha_{it} \cdot \bar{\mathbf{u}}_{it} \bar{\mathbf{u}}_{it}^\top\right)$, for $r$ reflections.
- **Kronecker (K):** $\mathbf{Q}_t = \mathcal{M}(\mathbf{x}_t) := \mathbf{I} - (\bar{\mathbf{K}}_t^1 \otimes \bar{\mathbf{K}}_t^2)$, where $\otimes$ denotes the Kronecker product.

This allows to reduce the size of the input-dependent parameters $\alpha_{it}$, $\bar{\mathbf{u}}_{it}$, and $\bar{\mathbf{K}}_t^i$ to $O(d)$, and consequently reduce the size of the linear map $\mathbf{W_Q}$ to $O(d^2 r)$ and $O(d^2)$. In order to guarantee stability, the condition $||\mathbf{I} - \mathbf{Q}||_2 < 1$ can be enforced by scaling $\alpha_{it}$, $\bar{\mathbf{u}}_{it}$, and $\bar{\mathbf{K}}_t^i$ appropriately. For more details about the channel mixer variants, please refer to App. C. Fig. 3b, we compare different channel mixer variants and observe that the Kronecker structure seems to be most appropriate the state-tracking task $A_5$.

## 3.4 Algorithmic Implications

Recall from Sec. 2 that an explicitly parametrized dense selective RNN can only be parallelized under strict assumptions on its structure and runs otherwise in $O(T)$ sequential steps. However, a parallelizable structure is given by the element-wise, diagonal transition $\mathbf{\Lambda}_t$ of a diagonal RNN (Martin & Cundy, 2018). Since such a diagonal RNN is called $\ell^*$-times as a subroutine of the fixed-point iteration in Eq. 6, a fixed-point RNN runs in $O(\ell^* \cdot \log T)$ sequential steps. This means that the implicit parametrization –as opposed to explicit or non-linear parametrizations– allows to decouple the number of sequential steps $\ell^*$ from the sequence length $T$ itself, and trade parallelism for expressivity.

This insight suggests an opportunity to introduce a non-linear computation for every sequential step, like in classical RNNs. Concretely, we investigate channel mixers $\mathcal{M}(\mathbf{x}_t + \mathbf{h}_{t-1}^{\ell-1})$ which are a function of both the input $\mathbf{x}_t$ and the hidden state $\mathbf{h}_{t-1}^{\ell-1}$ from the previous iteration (in both time and depth) without degrading parallelizability. In Fig. 3b, we compare channel mixers with and without hidden state dependence and observe that this indeed improves sequence length generalization.

Summarizing the results so far, we arrive at an updated recurrence with hidden state dependence:

$$\mathbf{h}_t^\ell = \boldsymbol{\lambda}_t^\ell \odot \mathbf{h}_{t-1}^\ell + (\mathbf{1} - \boldsymbol{\lambda}_t^\ell) \odot (\mathbf{Q}_t^\ell \mathbf{B}_t^\ell \mathbf{x}_t + (\mathbf{I} - \mathbf{Q}_t^\ell)\mathbf{h}_t^{\ell-1}), \tag{8}$$

where we use $\odot$ to highlight the parallelizability of the element-wise product. We would like to note that due to the normalization $(\mathbf{I} - \boldsymbol{\Lambda}_t)$, the corresponding dense RNN $F_{\boldsymbol{\theta}}$ is not explicitly representable anymore as discussed in App. B.2. Furthermore, for the time-varying parametrization in Eq. 8, the convergence guarantees may be weaker and solutions $\mathbf{h}^*$ could be non-unique due to the hidden state dependence. In practice, we iterate until $\frac{||\mathbf{h}^\ell - \mathbf{h}^{\ell-1}||_\infty}{||\mathbf{h}^\ell||_\infty} < 0.1$ and observe that the conditions of Thm. 3.1 are strong enough to reach convergence within a finite number of iterations $\ell^*$ as evidenced by Fig. 3c. Interestingly, the model navigates the *parallelism tradeoff* (Merrill & Sabharwal, 2023) and adaptively increases its sequential computation for harder tasks.

## 3.5 Optimizing Fixed-Point RNNs

One advantage of converging to a fixed-point as opposed to general layer looping lies in model training. Since the gradient with respect to $\mathbf{h}^0$ is not needed, implicit differentiation can be used to avoid storing and backpropagating through the computational graph of the fixed-point iteration, as discussed by Liao et al. (2018), Bai et al. (2019), and in App. B.3. In practice, truncated backpropagation of the last $k$ iterations suffices to approximate the gradient through the full iteration $\mathbf{J}_\mathbf{x}^* \approx \mathbf{J}_\mathbf{x}(\mathbf{h}^{\ell^*-k}) \cdot \ldots \cdot \mathbf{J}_\mathbf{x}(\mathbf{h}^{\ell^*})$. For Fixed-Point RNNs we observe that computing the gradient only at the fixed-point ($k = 0$), is enough to stabilize training. This means that compared to a single diagonal RNN layer, Fixed-Point RNNs incur no memory overhead and only sequential overhead in the forward pass but not in the backward pass.

We hypothesize that this is possible because $f_{\boldsymbol{\theta}}(\mathbf{x}, \mathbf{h})$ is a mostly linear object as opposed to multi-layer implicit models such as (Schöne et al., 2025). Furthermore, we observe that hidden state dependence $\mathcal{M}(\mathbf{x}_t + \mathbf{h}_{t-1}^{\ell-1})$ particularly helps with gradient-based optimization. We credit this to the symmetry between the gradients w.r.t. $\mathbf{x}$ and $\mathbf{h}$, and formalize this in the following theorem:

**Theorem 3.2.** *Let $f_{\boldsymbol{\theta}}(\mathbf{x}, \mathbf{h})$ have Lipschitz constant $< 1$ and fixed-point $\mathbf{h}^*$. If the Jacobians $\frac{\partial f_{\boldsymbol{\theta}}}{\partial \mathbf{x}}(\mathbf{x}, \mathbf{h})$ and $\frac{\partial f_{\boldsymbol{\theta}}}{\partial \mathbf{h}}(\mathbf{x}, \mathbf{h})$ are equal, then the gradient $\nabla_{\boldsymbol{\theta}} \mathcal{L}(f_{\boldsymbol{\theta}}(\mathbf{x}, \mathbf{h}), \mathbf{y})$ of the loss $\mathcal{L}(\cdot, \mathbf{y})$ for a target $\mathbf{y}$ at the fixed point $\mathbf{h} = \mathbf{h}^*$ is a descent direction of $\mathcal{L}(F_{\boldsymbol{\theta}}(\mathbf{x}), \mathbf{y})$. Proof in App. B.4.*

# 4 Fixed-Point Mamba

In the previous section we introduced the FP-RNN framework on a small RNN with vector hidden state. Now, we extend it to modern matrix state RNNs in Sec. 4.1 and parametrize a dense variant of Mamba (Gu & Dao, 2024) in Sec. 4.2. A detailed description of the architecture is available in App. C.2. We compare the architecture to the baselines Mamba (Gu & Dao, 2024), Mamba-2 (Dao & Gu, 2024), Gated DeltaNet (Yang et al., 2025), and LSTM (Hochreiter & Schmidhuber, 1997) on the copy task introduced by Jelassi et al. (2024) in Sec. 4.3 and state-tracking introduced by Merrill & Sabharwal (2023) in Sec. 4.4. In order to keep the number of layers at the same order of magnitude, we use two layers for the diagonal linear RNN baselines and one layer for FP-Mamba and LSTM. Finally, we discuss the required number of fixed-point iterations in the context of state-tracking and language modeling in Sec. 4.5.

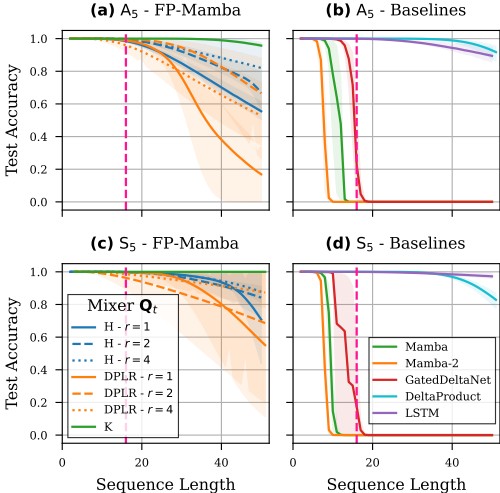

Figure 4: *Length generalization on $A_5$ (**a, c**) and $S_5$ (**b, d**) beyond the train sequence length 16 (pink line). We compare a 1-layer FP-Mamba with mixer variants $\mathbf{Q}_t$ to baselines with 2 layers.*

## 4.1 Introducing Matrix States

Memory capacity is an important consideration in RNNs. In preliminary experiments, we notice a clear gap between the performance of a Fixed-Point RNNs and Mamba in terms of copying ability. We attribute this difference in performance to Mamba's state-expansion which endows it with matrix hidden states similar to linear attention, DeltaNet, or mLSTM (Katharopoulos et al., 2020; Schlag et al., 2021a; Beck et al., 2024). In simple terms, these models use an outer product of an input-dependent vector $\mathbf{b}_t \in \mathbb{R}^{d_{\text{state}}}$ (i.e. the key) and the input vector $\mathbf{x}_t \in \mathbb{R}^{d_{\text{inner}}}$ (i.e. the value) as an input to a matrix-valued recurrence with hidden state and transition gate $\mathbf{H}_t, \boldsymbol{\lambda}_t \in \mathbb{R}^{d_{\text{state}} \times d_{\text{inner}}}$. The hidden state is then contracted with another input-dependent vector $\mathbf{c}_t \in \mathbb{R}^{d_{\text{state}}}$ (i.e. the query) to get the output $\mathbf{y}_t^\top = \mathbf{c}_t^\top \mathbf{H}_t \in \mathbb{R}^{d_{\text{inner}}}$:

$$\mathbf{H}_t = \boldsymbol{\lambda}_t \odot \mathbf{H}_{t-1} + \mathbf{b}_t \mathbf{x}_t^\top, \tag{9}$$

This matrix-valued recurrence introduces some challenges to our fixed-point framework. Specifically, in order to mix all the channels over the entirety of the state elements, the mixer has to be a fourth-order tensor $\mathcal{Q}_t \in \mathbb{R}^{d_{\text{state}} \times d_{\text{inner}} \times d_{\text{state}} \times d_{\text{inner}}}$ in

$$\mathbf{H}_t^\ell = \boldsymbol{\lambda}_t \odot \mathbf{H}_{t-1}^\ell + \mathcal{Q}_t \bullet \mathbf{b}_t \mathbf{x}_t^\top + (\mathcal{I} - \mathcal{Q}_t) \bullet \mathbf{H}_t^{\ell-1}, \tag{10}$$

where $\bullet$ denotes the tensor contraction $\text{einsum}(klij, ij \to kl)$ with fourth-order identity tensor $\mathcal{I}$ of the same shape as $\mathcal{Q}_t$. Certainly, computing the fixed-point introduced in Eq. 10 is very challenging both in terms of computation and memory. As we will confirm in Sec. 4.2, one solution is to pass the contracted output $y_t$ between fixed-point iterations

$$\mathbf{H}_t^\ell = \boldsymbol{\lambda}_t \odot \mathbf{H}_{t-1}^\ell + \mathbf{b}_t \left(\mathbf{Q}_t \mathbf{x}_t\right)^\top + \mathbf{b}_t \left((\mathbf{I} - \mathbf{Q}_t) \mathbf{y}_t^{\ell-1}\right)^\top. \tag{11}$$

This implicitly factorizes the tensor mixer $\mathcal{Q}_t$ into separately mixing along dimension $d_{\text{inner}}$ which is used for better expressivity, and dimension $d_{\text{state}}$ which is used for better memory capacity.

## 4.2 FP-Mamba Iteration

Let us apply the the fixed-point RNN framework to the Mamba parametrization. We represent the hidden state as $\mathbf{H}_t^\ell$, where $t$ is the token index (i.e., indexing over the sequence dimension), and $\ell$ is the fixed-point iteration index (i.e., indexing over the depth dimension). The same notation is used for other variables to emphasize when they depend on the input and hidden state of the current iteration. We propose the following iteration to adapt Mamba with notation from App. C.1 to the fixed-point mechanism for matrix state RNNs in Eq. 11:

$$\mathbf{H}_t^\ell = \boldsymbol{\lambda}_t \odot \mathbf{H}_{t-1}^\ell + \bar{\mathbf{b}}_t^\ell \left(\Delta_t \mathbf{Q}_t^\ell \mathbf{x}_t\right)^\top + \bar{\mathbf{b}}_t^\ell \left(\Delta_t \left(\mathbf{I} - \mathbf{Q}_t^\ell\right) \mathbf{y}_t^{\ell-1}\right)^\top,$$
$$\mathbf{y}_t^{\ell\top} = (\bar{\mathbf{c}}_t^\ell)^\top \mathbf{H}_t^\ell. \tag{12}$$

L2-normalizing $\bar{\mathbf{b}}_t^\ell$ and $\bar{\mathbf{c}}_t^\ell$ allows to limit the Lipschitz constant according to Theorem 3.1. Furthermore, we replace the normalization term $(1 - \boldsymbol{\lambda}_t)$ with Mamba's normalization term $\Delta_t$. Expanding $\mathbf{y}_t^{\ell-1}$ yields the recurrence on the matrix state

$$\mathbf{H}_t^\ell = \boldsymbol{\lambda}_t \odot \mathbf{H}_{t-1}^\ell + \bar{\mathbf{b}}_t^\ell (\Delta_t \mathbf{Q}_t^\ell \mathbf{x}_t)^\top + \bar{\mathbf{b}}_t^\ell (\bar{\mathbf{c}}_t^{\ell-1})^\top \mathbf{H}_t^{\ell-1} (\mathbf{I} - \mathbf{Q}_t^\ell)^\top \Delta_t, \tag{13}$$

where the last term nicely illustrates the two components which mix the channels of the hidden states: the low-rank matrix $\bar{\mathbf{b}}_t^\ell (\bar{\mathbf{c}}_t^{\ell-1})^\top$ mixes over the dimension $d_{\text{state}}$, while $(\mathbf{I} - \mathbf{Q}_t^\ell)^\top$ mixes over the dimension $d_{\text{inner}}$. This factorization significantly simplifies the fourth-order tensor mixer formulation introduced in Eq. 10, remains expressive as discussed in App. F, and performs well in practice.

Finally, Eq. 12 can be computed as Mamba with an adjusted input $\tilde{\mathbf{x}}_t^\ell = \mathbf{Q}_t^\ell \left(\mathbf{x}_t - \mathbf{y}_t^{\ell-1}\right) + \mathbf{y}_t^{\ell-1}$,

$$\mathbf{H}_t^\ell = \boldsymbol{\lambda}_t \odot \mathbf{H}_{t-1}^\ell + \bar{\mathbf{b}}_t^\ell \left(\Delta_t \tilde{\mathbf{x}}_t^\ell\right)^\top. \tag{14}$$

In other words, one fixed-point step consists of a channel mixing using $\mathbf{Q}_t$, followed by a sequence mixing using Mamba. This separation of concerns allows to speed up the parallel recurrence in time using the Mamba implementation. To find a fixed-point, the two phases are repeated until $\frac{\|\mathbf{y}^\ell - \mathbf{y}^{\ell-1}\|_\infty}{\|\mathbf{y}^\ell\|_\infty} < 0.1$ is satisfied. After these $\ell^*$ iterations, required for the model to converge to a fixed-point, $\mathbf{H}_t^*$ and $\mathbf{y}_t^*$ present the hidden state and output of the dense matrix-valued RNN $F_\theta$. Similar to Mamba, we apply a gated linear unit $\mathbf{g}_t \in \mathbb{R}^{d_{\text{inner}}}$ to the output, which we observe to provide a slight improvement in performance when present within the fixed-point loop: $\tilde{\mathbf{y}}_t^\ell = \mathbf{g}_t \odot \mathbf{y}_t^{\ell-1}$.

| Dependence on $\mathbf{y}_{t-1}^{\ell-1}$ | | | | Test Accuracy | |
|:---:|:---:|:---:|:---:|:---:|:---:|
| $\boldsymbol{\lambda}_t$ | $\mathbf{Q}_t$ | $\mathbf{b}_t$ | $\mathbf{c}_t$ | | |
| | | | | 0.11 | $\pm 0.00$ |
| ✓ | | | | 0.53 | $\pm 0.02$ |
| | ✓ | | | 0.45 | $\pm 0.05$ |
| ✓ | ✓ | | | 0.55 | $\pm 0.05$ |
| | | ✓ | ✓ | 0.81 | $\pm 0.01$ |
| ✓ | | ✓ | ✓ | 0.88 | $\pm 0.01$ |
| | ✓ | ✓ | ✓ | 0.86 | $\pm 0.02$ |
| ✓ | ✓ | ✓ | ✓ | 0.94 | $\pm 0.03$ |

Table 1: *Effect of shifted hidden state dependence $\mathbf{y}_{t-1}^{\ell-1}$ on copying at $\times 2$ length generalization. Each column determines which input-dependent component of the recurrence in Eq. 12 also depends on $\mathbf{y}_{t-1}^{\ell-1}$. Performance is unlocked by including a hidden dependence for $\mathbf{b}_t$ and $\mathbf{c}_t$.*

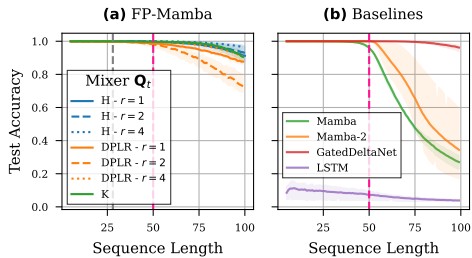

Figure 5: *Sequence length generalization on the copy task. A 1-layer FP-Mamba-H matches a 2-layer GatedDeltaNet baseline. Note that the median number of fixed-point iterations at test time $\ell^*$ (gray vertical line) is well below the longest training sequence length (pink line).*

## 4.3  Shifted Hidden State Dependence $\mathbf{y}_{t-1}^{\ell-1}$

In preliminary experiments, we observe that even the Fixed-Point RNN with input-dependent parameters and matrix state akin to Mamba-1 is outperformed by Mamba-2 or DeltaNet (Dao & Gu, 2024; Yang et al., 2024b) on a copy task. Inspired by the short convolution in Mamba, we investigate the effect of augmenting the input-dependence of parameters $\boldsymbol{\lambda}_t^\ell$, $\mathbf{b}_t^\ell$, $\mathbf{c}_t^\ell$, and $\mathbf{Q}_t^\ell$ at iteration $\ell$ with a shifted hidden state dependence. In practice, this means that these are linear functions of $\mathbf{x}_t$ as well as the shifted previous iterate in depth $\mathbf{y}_{t-1}^{\ell-1}$. We refer the reader to App. C.2 for the exact formulation of the dependency.

In Tab. 1, we ablate the hidden state dependence for various combinations of $\boldsymbol{\lambda}_t$, $\mathbf{b}_t$, $\mathbf{c}_t$, and a Householder $\mathbf{Q}_t$. Observe that the dependence of $\mathbf{b}_t$ and $\mathbf{c}_t$ is crucial to enable the model to copy. In App. C.4, we discuss why this dependence of $\mathbf{b}_t$ and $\mathbf{c}_t$ could be important for copying. If additionally $\boldsymbol{\lambda}_t$ and $\mathbf{Q}_t$ depend on $\mathbf{y}_{t-1}^{\ell-1}$, the copy task is essentially solvable at $\times 2$ length generalization. We therefore adopt the hidden state dependence for all components in FP-Mamba.

In Fig. 5, we evaluate length generalization on the copying task. While the best-performing baseline Gated DeltaNet is specifically designed for associative recall tasks (Yang et al., 2025), both Mamba 1 and 2 struggle with $\times 2$ generalization. FP-Mamba closes this gap and proves the effectiveness of our proposed modifications for better memory. We would like to highlight that the number of fixed-point iterations $\ell^*$ (gray vertical line) in FP-Mamba is well below the maximum sequence length.

## 4.4  State-Tracking

In Fig. 4, we evaluate the state-tracking capabilities of FP-Mamba with Kronecker, Householder, and DPLR channel mixers of $r \in \{1, 2, 4\}$ reflections or ranks, respectively. In particular, we compare our FP-Mamba to the baselines with regards to their length generalization beyond the training sequence length 16. As expected, LSTM solves $A_5$ and $S_5$, while Mamba and Mamba-2 are not able to learn it even at the training sequence length. Similar to Fig. 3b, the Kronecker structure seems to be the most suitable for the task. But FP-Mamba based on Householders also improves in terms of sequence length generalization presumably due to its improved memory. A comparison to the recent DeltaProduct (Siems et al., 2025) on training sequence length 128 is available in App. E.2.

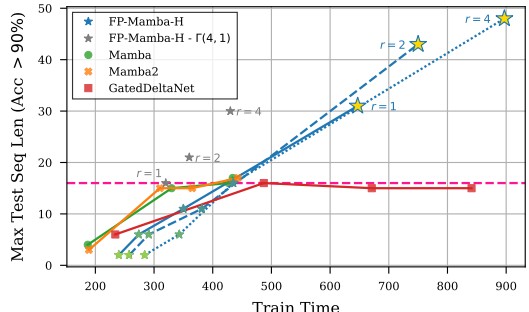

Figure 6: *Length generalization as a function of training time on $A_5$. Wall clock time is plotted against the longest test sequence length with $> 90\%$ accuracy for every model. While baselines of increasing depth cannot generalize beyond the training sequence length 16 (horizontal pink line), our proposed framework allows to achieve much higher generalization by scaling training time through the number of fixed-point iterations $\ell$.*

## 4.5 Required Number of Iterations $\ell^*$

A fixed-point iteration in the forward pass inevitably introduces sequential overhead to the computation of a model. While this might be acceptable for sequential generation at test time, reduced parallelism can be inhibiting at training time. In Fig. 1, we therefore evaluate FP-Mamba-H on $A_5$ with limited number of fixed-point iterations at training time $\ell_{\max} \in \{2, 4, 8, 16\}$. We observe that the performance decreases once $\ell_{\max}$ is lower than the training sequence length of 16. In Fig. 6, we confirm that the resulting longer training times are indeed required for good length generalization. However, as opposed to baselines of increasing depth $\in \{1, 2, 4, 6, 8\}$, fixed-point iterations gain from the additional training time. Furthermore, there is room to improve efficiency, as suggested by a simple randomization scheme

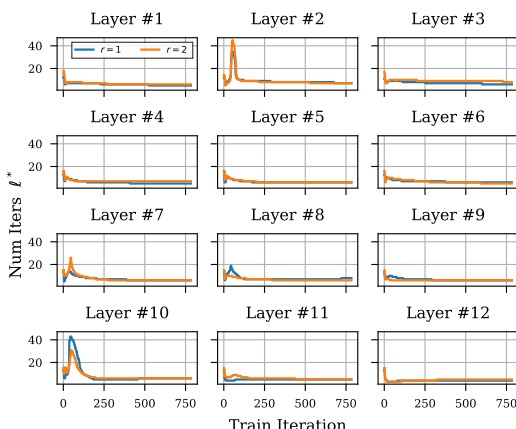

Figure 7: *The effective number of fixed-point iterations for each layer of a FP-Mamba-H throughout language pretraining on FineWeb (Penedo et al., 2024) at context length 2048. The corresponding validation perplexities are available in App. E.1.*

(gray stars) where $\ell_{\max} \sim \Gamma(4, 1)$ is sampled from a Gamma distribution with mean 4 for every batch. But most importantly, the effective number of fixed-point iterations depends on the difficulty of the task. Indeed, Fig. 7 shows that the model automatically adapts to using less fixed-point iterations on language pretraining at context length 2048. Similarly, on copying (Fig. 5) and and modular arithmetic (Fig. 10), we observe that the required number of fixed-point iterations $\ell^*$ is well below the sequence length $T$. This suggests that the model adapts to $O(T)$ complexity on simpler tasks when the full state-tracking expressivity is not required.

## 5 Discussion

A fixed-point mechanism, such as the one introduced in this paper, endows a parallelizable, diagonal linear RNN

|  | Forward | Backward |
|---|---|---|
| Mamba | $O(T)$ | $O(T)$ |
| FP-Mamba | $O\left((T + C_{\mathbf{Q}_t}) \cdot \min(\ell^*, \ell_{\max})\right)$ | $O(T + C_{\mathbf{Q}_t})$ |

Table 2: *Complexity of FP-Mamba in comparison to Mamba. The cost of channel mixing with structure $\mathbf{Q}_t$ is denoted by $C_{\mathbf{Q}_t}$.*

with the ability to dynamically increase the sequential computation and describe a dense linear RNN in the limit. Our results show that such a paradigm can enable both strong state-tracking and memory capabilities with a constant number of parameters in a combined sequence and channel mixing layer (Fig. 2). In fact, the fixed-point iteration gradually transforms a diagonal (i.e., channel-wise) RNN into a dense (i.e., channel-mixing) RNN, thereby allowing to trade parallel computation for expressivity (Fig. 1) without incurring additional cost during backpropagation (cf. Tab. 2).

For Fixed-Point RNNs to become competitive in practice, it is important to further understand the trade-offs between parallel and sequential computation. In the worst case, as shown in Tab. 2, FP-RNNs could behave like traditional, non-linear RNNs with quadratic runtime $O(T^2)$ if the sequential overhead $\ell^*$ is linear in the sequence length $T$. This, however, is not necessarily a disadvantage since FP-RNNs adapt $\ell^*$ to the difficulty of the task. In this paper, we focus on introducing the framework for FP-RNNs and leave the improvement of fixed-point convergence rates to future work.

Fixed-Point RNNs present an interesting opportunity to be fused into a single GPU kernel with reduced memory I/O. This is an inherent advantage from performing repeated computation on the same operands. Several open problems need to be solved to achieve that: (1) different implementations such as sequential, parallel, or chunk-wise should converge to the same fixed-points, (2) the memory footprint of the fixed-point iteration should satisfy current hardware limitations, and (3) alternative sequence or channel mixer structures could unlock higher efficiency. Future progress on these problems could enable significant speed-ups in practical implementations of Fixed-Point RNNs.

**Conclusion** In this paper, we presented a framework to cast a general class of dense linear RNNs as fixed-points of corresponding diagonal linear RNNs. Fixed-Point RNNs provide a mechanism to trade computation complexity for expressivity while uniting the expressivity of recurrent models with the improved memory of linear attention models. Following encouraging results on toy tasks specifically designed to assess these capabilities, we hope this paper enables more expressive sequence mixers.

## Acknowledgments and Disclosure of Funding

We would like to thank Riccardo Grazzi and Julien Siems for the helpful discussions and comments. Antonio Orvieto, Felix Sarnthein and Sajad Movahedi acknowledge the financial support of the Hector Foundation. Felix Sarnthein would also like to acknowledge the financial support from the Max Planck ETH Center for Learning Systems (CLS).

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

# Appendices

# A Background and Literature Review (Sec. 2)

Since their introduction (Rumelhart et al., 1986; Elman, 1990), RNNs have significantly contributed to the evolution of machine learning methods for sequential data, marked by key innovations such as the LSTM (Hochreiter & Schmidhuber, 1997) and Echo-State Networks (Jaeger, 2001). However, two significant challenges lead to the widespread adoption of the Transformer architecture (Vaswani et al., 2017): first, GPU hardware is optimized for large-scale matrix multiplications. Second, recurrent models are notoriously difficult to train due to vanishing and exploding gradients (Hochreiter et al., 2001; Pascanu et al., 2013).

**Beyond softmax attention.** The quadratic runtime complexity of Transformers motivated research on the linearization of its attention mechanism (Wang et al., 2020; Chen et al., 2021; Choromanski et al., 2020) – a technique that inevitably brings the sequence mixing mechanism closer to RNN-like processing (Katharopoulos et al., 2020; Schlag et al., 2021a). Recently, improvements on the long-range-arena benchmark (Tay et al., 2020) with state-space models (Gu et al., 2022; Smith et al., 2023) sparked a renewed interest in recurrent models (Gu & Dao, 2024; Sun et al., 2023; De et al., 2024; Qin et al., 2024; Peng et al., 2024; Yang et al., 2024a). New efficient token mixing strategies such as Mamba (Gu & Dao, 2024) showcase impressive results in language modeling (Waleffe et al., 2024) while offering linear runtime complexity. These models are fundamentally diagonal linear RNNs, which enables parallel algorithms such as parallel scans (Martin & Cundy, 2018) and fast linear attention based implementations (Yang et al., 2024b; Dao & Gu, 2024).

**Expressivity of Diagonal vs. Dense RNNs.** It was recently pointed out by Cirone et al. (2024b) that the diagonality in the hidden-to-hidden state transition inevitably causes expressivity issues, showcasing a stark distinction with classic dense nonlinear RNNs, known to be Turing-complete (Siegelmann & Sontag, 1992; Korsky, 2019) and fully expressive in a dynamical systems sense (Hanson & Raginsky, 2020). Merrill et al. (2024) pointed at a similar issue with diagonality using tools from circuit complexity: in contrast to e.g. LSTMs, diagonal linear RNNs can not express state-tracking algorithms. This issue sparked interest in designing fast non-diagonal recurrent mechanisms and, more generally, in providing architectures capable of solving state-tracking problems. The first example of such an architecture is DeltaNet (Yang et al., 2024b) employing a parallelizable Housholder reflection as a state transition matrix. Endowing this matrix with negative eigenvalues improves tracking in SSMs (Grazzi et al., 2024). In concurrent work, Siems et al. (2025) show that adding more reflections improves state-tracking.

**Toy tasks.** Several works propose toy tasks to identify specific shortcomings of modern architectures. Specifically, Beck et al. (2024) use the Chomsky hierarchy to organize formal language tasks, of which a modular arithmetic task remains unsolved. With similar motivations, Merrill & Sabharwal (2023) introduce a set of word-problems for assessing state-tracking capabilities, among which the $A_5$ and $S_5$ tasks remain unsolved by Transformers and SSMs. Motivated by Transformers outperforming RNNs in memory capabilities, Jelassi et al. (2024) introduce a copying task as a fundamental benchmark for memory. We focus on these tasks to evaluate our Fixed-Point RNN framework.

**Recurrence in Depth.** Machine learning models that reduce an intrinsic energy through iterations have been an object of interest for decades (Hopfield, 1982; Miyato et al., 2025). For example, recurrence in depth can increase the expressivity of Transformers (Dehghani et al., 2019; Schwarzschild et al., 2021; Giannou et al., 2023; Geiping et al., 2025) and is sometimes also understood as adaptive compute time (Graves, 2016). Under certain assumptions, iterated blocks can converge to an equilibrium point where they implicitly describe an expressive function (Bai et al., 2019; Ghaoui et al., 2021). Recently, this technique has been used to approximate non-linear RNNs with a fixed-point iteration of parallelizable linear RNNs (Lim et al., 2024; Gonzalez et al., 2024). In concurrent work to ours, Schöne et al. (2025) apply an iteration in depth to Mamba-2 and Llama blocks to increase expressivity and show promising results of their *implicit language models*. In contrast, we derive an explicit fixed-point iteration towards a dense linear RNN with a theoretically motivated parameterization, and focus on theoretical toy tasks.

# B  Fixed-Points as an RNN Layer (Sec. 3)

## B.1  Proof for Theorem 3.1 (Lipschitz constant of $f_\theta(\mathbf{x}, \mathbf{h})$ is $< 1$)

**Theorem 3.1.** *Let $f_{\boldsymbol\theta}(\mathbf{x}, \mathbf{h})$ be the diagonal linear RNN with input-independent $\boldsymbol\Lambda$ and $\mathbf{Q}$*

$$f_{\boldsymbol\theta} : (\mathbf{x}, \mathbf{h}) \mapsto \mathbf{h}', \qquad \mathbf{h}'_t = \boldsymbol\Lambda \mathbf{h}'_{t-1} + (\mathbf{I} - \boldsymbol\Lambda)\left(\mathbf{Q}\mathbf{B}_t\mathbf{x}_t + (\mathbf{I} - \mathbf{Q})\mathbf{h}_t\right). \qquad (7)$$

*If $\|\boldsymbol\Lambda\|_2 < 1$ and $\|\mathbf{I} - \mathbf{Q}\|_2 < 1$, then $f_{\boldsymbol\theta}(\mathbf{x}, \mathbf{h})$ has a Lipschitz constant $< 1$ in $\mathbf{h}$. Proof in App. B.1.*

We start the proof with the unrolled form of the linear RNN

$$f_{\boldsymbol\theta}(\mathbf{x}, \mathbf{h})_t = \sum_{\tau=0}^{t} \boldsymbol\Lambda^{t-\tau} (\mathbf{I} - \boldsymbol\Lambda)\left(\mathbf{Q}\mathbf{B}_\tau\mathbf{x}_\tau + (\mathbf{I} - \mathbf{Q})\mathbf{h}_\tau\right).$$

Note that in order to prove the theorem, we need to show that

$$\|f_{\boldsymbol\theta}(\mathbf{x}, \mathbf{h})_t - f_{\boldsymbol\theta}(\mathbf{x}, \mathbf{h}')_t\|_2 < \|\mathbf{h} - \mathbf{h}'\|_2,$$

where $\mathbf{h}$ and $\mathbf{h}'$ are two arbitrary hidden states. From the unrolled form, this is equivalent to

$$\left\| \sum_{\tau=0}^{t} \boldsymbol\Lambda^{t-\tau} (\mathbf{I} - \boldsymbol\Lambda)(\mathbf{I} - \mathbf{Q})(\mathbf{h}_\tau - \mathbf{h}'_\tau) \right\|_2 < \|\mathbf{h} - \mathbf{h}'\|_2. \qquad (15)$$

From the Cauchy-Schwarz inequality, we can upper-bound the LHS of Eq. 15 as

$$\left\| \sum_{\tau=0}^{t} \boldsymbol\Lambda^{t-\tau} (\mathbf{I} - \boldsymbol\Lambda)(\mathbf{I} - \mathbf{Q})(\mathbf{h}_\tau - \mathbf{h}'_\tau) \right\|_2 \leq \left\| \sum_{\tau=0}^{t} \boldsymbol\Lambda^{t-\tau} \right\|_2 \cdot \|\mathbf{I} - \boldsymbol\Lambda\|_2 \cdot \|\mathbf{I} - \mathbf{Q}\|_2 \cdot \left\| \mathbf{h}_{\leq t} - \mathbf{h}'_{\leq t} \right\|_2,$$

where $\mathbf{h}_{\leq t}$ corresponds to the concatenation of the hidden states $\mathbf{h}_\tau$ for $\tau \leq t$. Now to prove this product is $< \|\mathbf{h} - \mathbf{h}'\|_2$, consider the terms individually. Since $\left\| \mathbf{h}_{\leq t} - \mathbf{h}'_{\leq t} \right\|_2 \leq \|\mathbf{h} - \mathbf{h}'\|_2$, the remaining terms need to be $< 1$. Assuming $\boldsymbol\Lambda$ is contractive, we use the Neumann series $\sum_{\tau=0}^{t} \boldsymbol\Lambda^{t-\tau} \leq (\mathbf{I} - \boldsymbol\Lambda)^{-1}$ and get

$$\left\| \sum_{\tau=0}^{t} \boldsymbol\Lambda^{t-\tau} \right\|_2 \cdot \|\mathbf{I} - \boldsymbol\Lambda\|_2 \leq 1.$$

Finally, it remains to show that

$$\|\mathbf{I} - \mathbf{Q}\|_2 < 1.$$

This condition can be satisfied if $\mathbf{I} - \mathbf{Q}$ is contractive. This completes our proof. $\qquad\square$

## B.2  Effect of normalization factor $(\mathbf{I} - \boldsymbol\Lambda_t)$ on class of matrices $\mathbf{A}_t$

For the sake of exposition, the introduction of the implicit parametrization in Sec. 3.1 did not consider input normalization $(\mathbf{I} - \boldsymbol\Lambda_t)$. However, as discussed in Sec. 3.2 this is a crucial component to stabilize the recurrence in time. To derive the representable dense matrices $\mathbf{A}_t$ in the presence of the normalization factor $(\mathbf{I} - \boldsymbol\Lambda_t)$, let us start by assuming a fixed-point was found according to Thm. 3.2:

$$\mathbf{h}_t^* = \boldsymbol\Lambda_t \mathbf{h}_{t-1}^* + (\mathbf{I} - \boldsymbol\Lambda_t)(\mathbf{Q}_t \mathbf{B}_t \mathbf{x}_t + (\mathbf{I} - \mathbf{Q}_t)\mathbf{h}_t^*)$$
$$= \boldsymbol\Lambda_t \mathbf{h}_{t-1}^* + (\mathbf{I} - \boldsymbol\Lambda_t)\mathbf{Q}_t \mathbf{B}_t \mathbf{x}_t + (\mathbf{I} - \boldsymbol\Lambda_t)(\mathbf{I} - \mathbf{Q}_t)\mathbf{h}_t^*.$$

Rearranging the terms allows to move $\mathbf{h}_t^*$ to the other side

$$(\mathbf{I} - (\mathbf{I} - \boldsymbol\Lambda_t)(\mathbf{I} - \mathbf{Q}_t))\mathbf{h}_t^* = \boldsymbol\Lambda_t \mathbf{h}_{t-1}^* + (\mathbf{I} - \boldsymbol\Lambda_t)\mathbf{Q}_t \mathbf{B}_t \mathbf{x}_t.$$

Moving $(\mathbf{I} - (\mathbf{I} - \boldsymbol\Lambda_t)(\mathbf{I} - \mathbf{Q}_t))$ back to other side yields

$$\begin{aligned} \mathbf{A}_t &= (\mathbf{I} - (\mathbf{I} - \boldsymbol\Lambda_t)(\mathbf{I} - \mathbf{Q}_t))^{-1}\boldsymbol\Lambda_t \\ &= \left(\boldsymbol\Lambda_t^{-1}(\mathbf{I} - (\mathbf{I} - \boldsymbol\Lambda_t)(\mathbf{I} - \mathbf{Q}_t))\right)^{-1} \\ &= \left(\boldsymbol\Lambda_t^{-1} - (\boldsymbol\Lambda_t^{-1} - \mathbf{I})(\mathbf{I} - \mathbf{Q}_t)\right)^{-1} \\ &= \left(\mathbf{I} + (\boldsymbol\Lambda_t^{-1} - \mathbf{I})\mathbf{Q}_t\right)^{-1} \end{aligned}$$

Following the standard assumptions that $\mathbf{0} \preceq \mathbf{\Lambda}_t, \mathbf{Q}_t \preceq \mathbf{I}$, the matrix $(\mathbf{I} + (\mathbf{\Lambda}_t^{-1} - \mathbf{I})\mathbf{Q}_t)$ is full rank and $\succeq \mathbf{I}$. Therefore its inverse $\mathbf{A}_t$ exists and is contractive. The expressivity of $\mathbf{A}_t$ is only limited if $\mathbf{\Lambda}_t \approx \mathbf{I}$. This however would also be problematic for diagonal SSM and therefore the Mamba initialization is bias towards $\mathbf{\Lambda}_t \prec \mathbf{I}$. Thus, the normalization does not pose a significant problem for the expressivity of $\mathbf{A}_t$ in practice.

## B.3   Implicit Differentiation for Optimizing Fixed-Point RNNs

One advantage of converging to a fixed-point over general layer looping lies in model training. Since the gradient with respect to $\mathbf{h}^0$ is not needed, implicit differentiation can be used to avoid storing and backpropagating through the computational graph of the fixed-point iteration, as discussed by Liao et al. (2018), Bai et al. (2019). To see this, consider the Jacobian across $\ell$ iterations $\mathbf{J}_{\mathbf{x}}^{\ell} = \frac{\partial f_{\boldsymbol{\theta}}}{\partial \mathbf{x}}(\mathbf{x}, \mathbf{h}^{\ell-1})$. Since $\mathbf{h}^{\ell-1}$ depends on $\mathbf{x}$ as well, we can recursively express $\mathbf{J}_{\mathbf{x}}^{\ell}$ in terms of $\mathbf{J}_{\mathbf{x}}^{\ell-1}$ and the Jacobians of a single iteration $\mathbf{J}_{\mathbf{x}}(\mathbf{h}) = \frac{\partial f_{\boldsymbol{\theta}}}{\partial \mathbf{x}}(\mathbf{x}, \mathbf{h})$ and $\mathbf{J}_{\mathbf{h}} = \frac{\partial f_{\boldsymbol{\theta}}}{\partial \mathbf{h}}(\mathbf{x}, \mathbf{h})$ by applying the chain rule

$$\mathbf{J}_{\mathbf{x}}^{\ell} = \mathbf{J}_{\mathbf{x}}(\mathbf{h}^{\ell-1}) + \mathbf{J}_{\mathbf{h}^{\ell-1}} \cdot \mathbf{J}_{\mathbf{x}}^{\ell-1}. \tag{16}$$

Instead of unrolling, we can implicitly differentiate $\mathbf{h}^* = f_{\boldsymbol{\theta}}(\mathbf{x}, \mathbf{h}^*)$ w.r.t. $\mathbf{x}$, which yields $\mathbf{J}_{\mathbf{x}}^* = \mathbf{J}_{\mathbf{x}}(\mathbf{h}^*) + \mathbf{J}_{\mathbf{h}^*} \cdot \mathbf{J}_{\mathbf{x}}^*$. Given the conditions on the Lipschitz constant of $f_{\boldsymbol{\theta}}(\mathbf{x}, \mathbf{h})$ in $\mathbf{h}$, we can assume $\mathbf{J}_{\mathbf{h}^{\ell}}$ to be contractive and therefore $(\mathbf{I} - \mathbf{J}_{\mathbf{h}^{\ell}})$ to be positive definite and invertible. This allows to reformulate as

$$\mathbf{J}_{\mathbf{x}}^* = (\mathbf{I} - \mathbf{J}_{\mathbf{h}^*})^{-1} \cdot \mathbf{J}_{\mathbf{x}}(\mathbf{h}^*). \tag{17}$$

The case for $\mathbf{J}_{\boldsymbol{\theta}}^*$ works analogously. This means that the gradient w.r.t. the input $x$ and parameters $\boldsymbol{\theta}$ can be computed at the fixed-point with the cost of solving $(\mathbf{I} - \mathbf{J}_{\mathbf{h}^*})^{-1}$. Bai et al. (2021) and Schöne et al. (2025) approximate this inverse using the first terms of the Neumann series, which leads to a truncated backpropagation formulation or *phantom gradients*, incurring sequential overhead. For iteration with hidden state dependence, we can avoid this inversion altogether with Thm. 3.2:

**Theorem 3.2.** *Let $f_{\boldsymbol{\theta}}(\mathbf{x}, \mathbf{h})$ have Lipschitz constant $< 1$ and fixed-point $\mathbf{h}^*$. If the Jacobians $\frac{\partial f_{\boldsymbol{\theta}}}{\partial \mathbf{x}}(\mathbf{x}, \mathbf{h})$ and $\frac{\partial f_{\boldsymbol{\theta}}}{\partial \mathbf{h}}(\mathbf{x}, \mathbf{h})$ are equal, then the gradient $\nabla_{\boldsymbol{\theta}} \mathcal{L}(f_{\boldsymbol{\theta}}(\mathbf{x}, \mathbf{h}), \mathbf{y})$ of the loss $\mathcal{L}(\cdot, \mathbf{y})$ for a target $\mathbf{y}$ at the fixed point $\mathbf{h} = \mathbf{h}^*$ is a descent direction of $\mathcal{L}(F_{\boldsymbol{\theta}}(\mathbf{x}), \mathbf{y})$. Proof in App. B.4.*

In simple terms, Thm. 3.2 shows that parameterizing $f_{\boldsymbol{\theta}}(\mathbf{x}, \mathbf{h})$ such that $\mathbf{J}_{\mathbf{x}}(\mathbf{h}) = \mathbf{J}_{\mathbf{h}}$ guarantees optimization progress even if the gradient is computed only at the fixed-point. In practice, we observe that adhering to this condition in the form of hidden state dependence speeds-up the convergence of the model during training.

## B.4   Proof for Theorem 3.2 (Gradient of $f_{\theta}(\mathbf{x}, \mathbf{h})$ is a descent direction of $F_{\theta}(\mathbf{x})$)

We start the proof by setting $\boldsymbol{\delta} := \frac{\partial \mathcal{L}}{\partial f}$ and $\mathbf{J}_{\mathbf{x}} := \mathbf{J}_{\mathbf{x}}(\mathbf{h}^*)$. Then, we can write the backward propagation as $\frac{\partial \mathcal{L}}{\partial \mathbf{x}} = (\mathbf{J}_{\mathbf{x}}^*)^{\top} \boldsymbol{\delta}$. In order to prove that the gradient computed at the fixed-point is a descent direction, we need to show that $\mathbf{J}_{\mathbf{x}}^{\top} \boldsymbol{\delta}$ is in the direction of $(\mathbf{J}_{\mathbf{x}}^*)^{\top} \boldsymbol{\delta}$, or in other words, we have $\boldsymbol{\delta}^{\top} \mathbf{J}_{\mathbf{x}}^* \mathbf{J}_{\mathbf{x}}^{\top} \boldsymbol{\delta} \geq 0$. This is equivalent to showing that the symmetric part of the matrix $\mathbf{J}_{\mathbf{x}}^* \mathbf{J}_{\mathbf{x}}^{\top}$ is positive semi-definite.

Now note that from Eq. 17 we have: $\mathbf{J}_{\mathbf{x}}^* \mathbf{J}_{\mathbf{x}}^{\top} = (\mathbf{I} - \mathbf{J}_{\mathbf{h}})^{-1} \mathbf{J}_{\mathbf{x}} \mathbf{J}_{\mathbf{x}}^{\top}$. From our assumption $\mathbf{J}_{\mathbf{x}} = \mathbf{J}_{\mathbf{h}} := \mathbf{J}$, we need to show that the symmetric part of the matrix $(\mathbf{I} - \mathbf{J})^{-1} \mathbf{J} \mathbf{J}^{\top}$ is positive semi-definite. Note that $(\mathbf{I} - \mathbf{J})^{-1}$ and $\mathbf{J}$ commute by application of the Neumann series

$$(\mathbf{I} - \mathbf{J})^{-1} \mathbf{J} = \sum_{i=1}^{\infty} \mathbf{J}^i = \mathbf{J} \sum_{i=0}^{\infty} \mathbf{J}^i = \mathbf{J} (\mathbf{I} - \mathbf{J})^{-1},$$

which yields $(\mathbf{I} - \mathbf{J})^{-1} \mathbf{J} \mathbf{J}^{\top} = \mathbf{J} (\mathbf{I} - \mathbf{J})^{-1} \mathbf{J}^{\top}$. Going back to the definition of positive semi-definiteness, we need to show that $\boldsymbol{\delta}^{\top} \mathbf{J} (\mathbf{I} - \mathbf{J})^{-1} \mathbf{J}^{\top} \boldsymbol{\delta} > 0$ for all $\boldsymbol{\delta}$. Setting $\boldsymbol{\omega} = \mathbf{J}^{\top} \boldsymbol{\delta}$, this is equivalent to having $\boldsymbol{\omega}^{\top} (\mathbf{I} - \mathbf{J})^{-1} \boldsymbol{\omega}$. Note that from our assumption for the Lipschitz constant of the function, we have $\|\mathbf{J}\|_2 < 1$, which means $(\mathbf{I} - \mathbf{J})$ and $(\mathbf{I} - \mathbf{J})^{-1}$ have strictly positive eigenvalues. This completes our proof. $\square$

## C Fixed-Point Mamba (Sec. 4)

### C.1 Mamba: Selective SSMs

Mamba is a multi-layer network, with an embedding size of $d_{\text{model}}$. A Mamba block is a matrix state diagonal linear RNN which first expands a sequence of embeddings by a factor of $e$ to size $d_{\text{inner}} = e \times d_{\text{model}}$, and then computes an element-wise recurrence on the matrix hidden states $\mathbf{H}_t \in \mathbb{R}^{d_{\text{state}} \times d_{\text{inner}}}$ as

$$\mathbf{H}_t = \boldsymbol{\lambda}_t \odot \mathbf{H}_{t-1} + \mathbf{b}_t \left(\Delta_t \mathbf{x}_t\right)^\top, \tag{18}$$

where $\boldsymbol{\lambda}_t \in \mathbb{R}^{d_{\text{state}} \times d_{\text{inner}}}$ is an input-dependent state transition vector, $\mathbf{b}_t \in \mathbb{R}^{d_{\text{state}}}$ an input transition vector, $\mathbf{x}_t \in \mathbb{R}^{d_{\text{inner}}}$ the input, and $\Delta_t \in \mathbb{R}^{d_{\text{inner}} \times d_{\text{inner}}}$ a diagonal matrix which acts an input normalization term. The matrices are parameterized as:

$$\boldsymbol{\lambda}_t = \exp\left(-\boldsymbol{\lambda}_{\log} \Delta_t\right), \qquad\qquad \boldsymbol{\lambda}_{\log} = \exp\left(\boldsymbol{\omega}\right),$$
$$\Delta_t = \text{diag}\left(\text{softplus}\left(\mathbf{W}_\Delta \mathbf{x}_t + b_\Delta\right)\right), \qquad\qquad \mathbf{b}_t = \mathbf{W}_\mathbf{b} \mathbf{x}_t,$$

with $\boldsymbol{\omega} \in \mathbb{R}^{d_{\text{state}} \times d_{\text{inner}}}$, $\mathbf{W}_\Delta \in \mathbb{R}^{d_{\text{inner}} \times d_{\text{inner}}}$, $\mathbf{W}_\mathbf{b} \in \mathbb{R}^{d_{\text{state}} \times d_{\text{inner}}}$, and $b_\Delta \in \mathbb{R}^{d_{\text{inner}}}$. The output of a Mamba block $\mathbf{y}_t \in \mathbb{R}^{d_{\text{inner}}}$ is a contraction of the matrix hidden state with $\mathbf{c}_t \in \mathbb{R}^{d_{\text{state}}}$

$$\mathbf{y}_t^\top = \mathbf{c}_t^\top \mathbf{H}_t, \quad \mathbf{c}_t = \mathbf{W}_\mathbf{c} \mathbf{x}_t,$$

for $\mathbf{W}_\mathbf{c} \in \mathbb{R}^{d_{\text{state}} \times d_{\text{inner}}}$. Note that Mamba proposes a skip connection of $\mathbf{y}_t + \mathbf{D} \odot \mathbf{x}_t$, where $\mathbf{D} \in \mathbb{R}^{d_{\text{inner}}}$ is an input-independent vector. Finally, the model output is usually scaled by a gated linear unit (GLU) as $\tilde{\mathbf{y}}_t = \mathbf{g}_t \odot \mathbf{y}_t$, where $\mathbf{g}_t = \text{SiLU}\left(\mathbf{W}_\mathbf{g} \mathbf{x}_t\right)$ is a non-linear function of the input.

### C.2 FP-Mamba Parametrization

In our design of FP-Mamba, we aim to minimize our interventions in the underlying architecture in order to showcase the adaptability of our proposed framework. Consequently, we do not modify the careful parameterization of $\boldsymbol{\lambda}$ and the weight-tied normalization factor $\Delta_t$ proposed in the original Mamba formulation, and instead rely on layer normalization to limit the Lipschitz constant of the Mamba function. Specifically, in the FP-Mamba model we redefine $\mathbf{b}_t$ and $\mathbf{c}_t$ as $\mathbf{b}_t^\ell = \mathbf{W}_\mathbf{b}^\mathbf{y} \mathbf{y}_{t-1}^{\ell-1} + \mathbf{W}_\mathbf{b}^\mathbf{x} \mathbf{x}_t$ and $\mathbf{c}_t = \mathbf{W}_\mathbf{c}^\mathbf{y} \mathbf{y}_{t-1}^{\ell-1} + \mathbf{W}_\mathbf{c}^\mathbf{x} \mathbf{x}_t$. The remaining components, namely the state transition matrix $\boldsymbol{\lambda}_t$ and the GLU component are parameterized identically to Mamba.

The normalization is applied to the output of the model $\mathbf{y}_t$ after each iteration. While in theory projecting the output onto the unit sphere does not guarantee a Lipschitz constant $< 1$, we observe that in practice, this helps with stabilizing the forward and backward pass of the fixed-point RNN framework. We attribute this observation to the fact that achieving a $> 1$ Lipschitz constant requires the output of the RNN to become its additive inverse after an iteration, which rarely happens in practice.

### C.3 Parameterizing the mixers

We parameterize the channel mixer variants as follows:

- **Diagonal Plus Low Rank:** we define $\mathbf{u}_{it}^\ell = \text{SiLU}\left(\mathbf{W}_{\mathbf{u}_i}^\mathbf{x} \mathbf{x}_t + \mathbf{W}_{\mathbf{u}_i}^\mathbf{y} \mathbf{y}_{t-1}^{\ell-1}\right)$ and $\alpha_{it} = \sigma\left(\left(\mathbf{w}_{\alpha_i}^\mathbf{x}\right)^\top \mathbf{x}_t + \left(\mathbf{w}_{\alpha_i}^\mathbf{y}\right)^\top \mathbf{y}_{t-1}^{\ell-1} + b_{\alpha_i}\right)$, where SiLU(.) and $\sigma(.)$ are the SiLU and the sigmoid functions, respectively.

- **Householder Reflections:** we define similar to the diagonal plus low-rank variant.

- **Kronecker:** we define $\mathbf{D}_t^{\ell,n} = \text{diag}\left(\sigma\left(\mathbf{W}_{\mathbf{D}^n}^\mathbf{x} \mathbf{x}_t + \mathbf{W}_{\mathbf{D}^n}^\mathbf{y} \mathbf{y}_{t-1}^{\ell-1} + b_{\mathbf{D}^n}\right)\right)$ and $\mathbf{K}_t^n = \text{mat}\left(\text{SiLU}\left(\mathbf{W}_{\mathbf{K}^n}^\mathbf{x} \mathbf{x}_t + \mathbf{W}_{\mathbf{K}^n}^\mathbf{y} \mathbf{y}_{t-1}^{\ell-1} + b_{\mathbf{K}^n}\right)\right)$ for $n = 1, 2$, where diag(.) is the operator transforming a vector into a diagonal matrix, and mat(.) is the operator transforming a size $d$ vector into a $\sqrt{d} \times \sqrt{d}$ matrix.

For the diagonal plus low rank and the Householder reflections mixers, we L2 normalize the vectors $\mathbf{u}_{it}$ to achieve the unit vector formulation. Note that this does not guarantee a contractive diagonal plus low rank structure, which is why the first variant of the channel mixers are excluded form our

FP-RNN experiments. For the Kronecker variant, we define the matrices $\mathbf{K}_t^n$ as symmetric and positive semi-definite using the Cholesky decompositon structure, and normalize them by their largest eigenvalues. The largest eigenvalue is found using the power iterations method, which we found to be much more efficient for small-scale matrices compared to the functions in the PyTorch framework provided for this purpose.

In all of these parameterization, computing a matrix vector product for each fixed-point iteration can be performed in subquadratic time. Specifically, for the DPLR and the Householder formulation, the computation can be performed in linear time in state-size, while in the kronecker product variant, it can be performed in $\sqrt{d} \times \sqrt{d}$ for $d$ state-size.

## C.4 Dependence on $\mathbf{H}_{t-1}$ in theory

We hypothesize that the dependence of the matrices $\boldsymbol{\lambda}_t$, $\mathbf{b}_t$, $\mathbf{c}_t$, and $\mathbf{Q}_t$ may provide a mechanism for the model to retain and manipulate positional information over the sequence. Jelassi et al. (2024) and Trockman et al. (2024) show that position embeddings could play a crucial role in copy tasks by acting similar to hashing keys in a hashing table. We extend their mechanistic approach to understand why two-layers of linear attention could need $\mathbf{H}_{t-1}^{\ell-1}$ to generate appropriate position embeddings for the hashing mechanism.

Specifically consider $\mathbf{y}_t^\top = \mathbf{c}_t^\top \mathbf{H}_t$ with $\mathbf{H}_t = \mathbf{H}_{t-1} + \mathbf{b}_t \mathbf{x}_t^\top$, assuming that a linear RNN with matrix-state can express linear attention by setting $\boldsymbol{\lambda}_t \approx \mathbf{1} \ \forall t$. Upon receiving an input sequence $\{\mathbf{x}_1, \mathbf{x}_2, \ldots, \mathbf{x}_\delta\}$ of length $\delta$ followed by a delimiter element $\mathbf{x}_\mathbf{s}$, the model is expected to copy the input sequence autoregressively, i.e. to start producing $\{\mathbf{x}_1, \mathbf{x}_2, \ldots, \mathbf{x}_\delta\}$ at output positions $\delta + 1$ to $2\delta$. Following Arora et al. (2024), the second layer could use position embeddings as hashing keys to detect and copy each token. More concretely, if the first layer receives a sequence $\{\mathbf{x}_1, \mathbf{x}_2, \ldots, \mathbf{x}_\delta, \mathbf{x}_\mathbf{s}, \mathbf{x}_1, \mathbf{x}_2, \ldots, \mathbf{x}_{\delta-1}\}$ of size $2\delta$ and augments it with shifted position embeddings $\{\mathbf{p}_i\}_{i=1}^\delta$ to produce the hidden sequence $\{\mathbf{x}_1 + \mathbf{p}_1, \mathbf{x}_2 + \mathbf{p}_2, \ldots, \mathbf{x}_\delta + \mathbf{p}_\delta, \mathbf{x}_\mathbf{s} + \mathbf{p}_1, \mathbf{x}_1 + \mathbf{p}_2, \ldots, \mathbf{x}_{\delta-1} + \mathbf{p}_\delta\}$, then a second layer can act as a linear transformer and produce the sequence $\{\mathbf{x}_1, \mathbf{x}_2, \ldots, \mathbf{x}_\delta\}$ at output positions $\delta + 1$ to $2\delta$. In the following, we focus on the conditions for the first layer to produce the shifted position embeddings.

We start by assuming that the first layer has a skip-connection $\mathbf{y}_t^\top = \mathbf{c}_t^\top \mathbf{H}_t + \mathbf{x}_t^\top$. In this case, the model can augment the inputs with positional embeddings $\{\mathbf{p}_i\}_{i=1}^\delta$ if it is able to produce shifted encodings $\mathbf{p}_{t-\delta} = \mathbf{p}_t$ for $\delta < t$ using $\mathbf{p}_t^\top = \mathbf{c}_t^\top \mathbf{H}_t$. This condition can be unrolled as

$$\mathbf{p}_{t-\delta}^\top = \mathbf{c}_{t-\delta}^\top \mathbf{H}_{t-\delta} \overset{!}{=} \mathbf{c}_t^\top \mathbf{H}_{t-\delta} + \mathbf{c}_t^\top \sum_{\tau=t-\delta+1}^{t} \mathbf{b}_\tau \mathbf{x}_\tau^\top = \mathbf{p}_t^\top \qquad \forall \delta < t.$$

and is satisfied if the equations

$$\mathbf{c}_{t-\delta}^\top \mathbf{H}_{t-\delta} \overset{!}{=} \mathbf{c}_t^\top \mathbf{H}_{t-\delta} \qquad \text{and} \qquad \mathbf{c}_t^\top \sum_{\tau=t-\delta+1}^{t-1} \mathbf{b}_\tau \mathbf{x}_\tau^\top \overset{!}{=} -\mathbf{c}_t^\top \mathbf{b}_t \mathbf{x}_t^\top$$

hold. Such conditions could only be true if $\mathbf{b}_t$ and $\mathbf{c}_t$ are a function of the previous hidden state $\mathbf{H}_{t-1}$ because they need to be able to retain information about $\{\mathbf{x}_i\}_{i=t-\delta+1}^{t-1}$. While not an explicit mechanism for copying, this derivation provides insight into why a dependency on $\mathbf{H}_{t-1}$ could be helpful.

# D    Evaluation

## D.1    Task Descriptions

In this section, we provide task descriptions for the tasks used in the main text.

**State Tracking**    The task of tracking state in the alternating group on five elements ($A_5$) is one of the tasks introduced in (Merrill et al., 2024) to show that linear RNNs and SSMs cannot solve state-tracking problems. $A_5$ is the simplest subset of $S_5$, the word problem involving tracking the permutation of five elements. In these tasks, a model is presented with an initial state and a sequence of permutations. As the output, the model is expected to predict the state that results from applying the permutations to the initial state. Solving these task with an RNN requires either a dense transition matrix or the presence of non-linearity in the recurrence. It is therefore a good proxy to verify the state-tracking ability of FP-Mamba. In order to investigate the out-of-distribution generalization ability of the model, we train the model with a smaller train sequence length and evaluate for larger (more than $\times 3$) sequence lengths.

**Copying**    We use the copy task (Jelassi et al., 2024) in order to assess the memory capabilities of FP-Mamba. In this task, the model is presented with a fixed-size sequence of elements, and expected to copy a subsequence of it after receiving a special token signaling the start of the copying process. In order to investigate the out-of-distribution generalization ability of the model, we train the models with sequence length $< 50$, and assess the $\times 2$ length generalization following Jelassi et al. (2024) and Trockman et al. (2024).

## D.2    Experimental Details

In this section, we will provide our experiment setup for the state tracking, copying, and mod arithmetic tasks. The code is available at github.com/dr-faustus/fp-rnn.

**State tracking.**    We train all models for 5 epochs, with a batch size of 512, 3 different random seeds, learning rate set to 0.0001, weight decay set to 0.01, gradient clipping 1.0, and the AdamW optimizer (Loshchilov & Hutter, 2017). For the train data, we sample 16M datapoints from all the possible permutations for a sequence length of 16, and split the data with a ratio of 4 to 1 for train and validation samples. For the test data, we sample 500k sequences of length 50. We use the implementation and the hyperparameters provided by Merrill et al. (2024) both for data generation and train/test. We train the model for sequence length 16 on the train sample, and evaluate for sequence lengths 2 through 50 on the test sample. Consequently, each epoch of training consists of 25428 iterations, making the total number of iterations during training to be around 1.25M. Note that the likelihood of overlap between the train and test samples is negligible since exhaustive generation of samples in $S_5$ and $A_5$ at sequence length $k$ would amount to $60^k$ and $30^k$, respectively.

**Copying.**    We train all models for 10000 iterations, batch size 128, 3 different random seeds, learning rate 0.00001, weight decay 0.1, gradient clipping 1.0, the AdamW optimizer, and with linear learning rate decay after a 300 iterations warmup. The data is sampled randomly at the start of the training/evaluation. We use a vocab size of 29, a context length of 256, and train the model for copy sequence length in the range 5 to 50, and evaluate for the range 5 to 100. we use the implementation and the hyperparameters provided by Jelassi et al. (2024).

**Mod arithmetic.**    Our models are trained for 100000 iterations, batch size 256, learning rate 0.001, weight decay 0.1, and no gradient clipping. The learning rate is decayed using a cosine scheduling by a factor of 0.001 after 10000 iterations of warmup. The data is randomly sampled at the start of training/evaluation. We use a vocab size of 12, with context length 256, and train data sequence length in the range 3 to 40, and the test/evaluation data in the range 40 to 256. We use the implementation and the hyperparameters provided by Beck et al. (2024) and Grazzi et al. (2024), which are the same hyperparameters used for training and evaluating the baselines.

**Language Modeling.**    For the language modeling task, we use the implmentation provided by Ajroldi (2024). We use a batchsize of $16 \times 4 \times 4 = 256$, training on 4 A100-80GB GPUs with 4 accumulation steps, which is the batchsize used in the 2.5B setting in (Gu & Dao, 2024). The

learning rate is optimized for the Mamba model (0.004) and train all models with this learning rate, with cosine warmup with 0.1 steps. We use the AdamW optimizer with weight decay set to 0.1 and $\beta_1, \beta_2$ set to $0.9, 0.95$.

**Training Time on $A_5$.** In order to compare the proposed model to the baselines in terms of computation time, we train all of the baselines and our proposed model using the same hardware (A100-80GB gpus) on the $A_5$ task. We present the results in Fig. 6. Our Fixed-Point Mamba is trained at different maximum number of fixed-point iterations: between 2 (green) and 16 (blue), or sampled from the Gamma distribution $\Gamma(4, 1)$ with mean 4 (gray).

**catbAbI** In this experiment, we use the setting provided by Schlag et al. (2021b). We optimize the learning rates on Mamba, and use the same learning rate to train FP-Mamba, which we found to be $5 \times 10^{-4}$. We use a batch size of 256, along with short convolutions, and 1, 2, or 4 layers. We set the maximum number of iterations $\ell_{\max}$ to 100.

### D.3 Heuristics to reduce the number of fixed-point iterations

Given the importance of scalability in current machine learning research, an implicit network needs to be as efficiently designed and implemented as possible. While our theoretical framework improves upon the memory and computational requirements on the backward pass, the forward, and especially finding the fixed-point through fixed-point iterations needs further consideration. In our preliminary experiments, we discover two heuristics that can help with improving this aspect significantly.

The first heuristic is relaxing our definition of convergence to the fixed-point during training. We observe that the number of iterations required to find the fixed-point for the sequences in the model usually has a power-law distribution, with certain outliers in each batch elongating the convergence time. In our experiments, we notice very little difference in the performance of the converged model when we exclude these sequences from our stopping criterion. Consequently, during training, we continue the fixed-point iterations procedure until a certain percentage of the datapoints in the batch (usually set to 75%) satisfy our criteria for convergence.

The second heuristic involves using a momentum-like update rule to accelerate the convergence of fixed-point iterations for certain sequences. Specifically, we observe that by setting the fixed-point update rule to $\mathbf{h}^{\ell+1} = \delta \cdot f_{\boldsymbol{\theta}}(\mathbf{x}, \mathbf{h}^\ell) + (1 - \delta) \cdot \mathbf{h}^\ell$ for some $\delta \in [0, 1]$, we can accelerate the convergence for certain sequences that are particularly slow to converge. Since this update rule can result in a biased approximation of the fixed-point, we implement a patience-based system that starts with $\delta = 1$, and reduces the value of $\delta$ exponentially when the residues fail to improve.

# E Additional Experimental Results

## E.1 Language Modeling

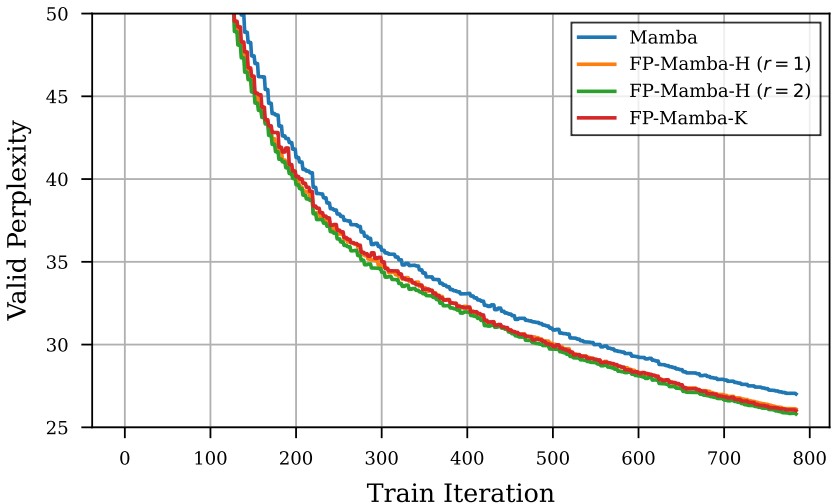

Figure 8: *The validation perplexity of the Mamba model vs. FP-Mamba-K and FP-Mamba-H with $r \in \{1, 2\}$ reflections. Note that all of the hyperparameters of the models are identical for fair comparison.*

In order to confirm the utility of the fixed-point framework in non-state-tracking settings, we performed an experiment on language modeling. Specifically, we compare the performance of a Mamba with an FP-Mamba, with the same hidden size (768) and number of layers (12). The settings are selected according to the 2.5B setup introduced in Gu & Dao (2024). We use a train subsample of the FineWeb dataset (Penedo et al., 2024) with 2B tokens, and a validation subsample with 200K tokens. We use a context length of 2048 for our experiment. For the FP-Mamba model, we use the Householder mixer with 1 and 2 reflections. We report the validation perplexity in Fig. 8.

As we can observe, the fixed-point framework does introduce a significant improvement to the performance of the model on perplexity. However, we note that this improvement cannot be only attributed to the multi-layer hypothesis of implicit models (Giannou et al., 2023), as increasing the number of Householder reflections does seem to be improving the perplexity further. Furthermore, we point out the practicality of the setup, as we can observe in Fig. 7 that in the absence of a state-tracking problem, the number of fixed-point iterations seems to be independent of the sequence length, and instead hover in the $< 10$ range. Finally, fixed-point iterations are not required in the backward bass and therefore only increase training time moderately.

## E.2 Long-Range State-Tracking

In this section, we investigate the ability of our proposed method in doing state tracking on longer sequences. Specifically, we will use the $A_5$ and $S_5$ datasets and train on sequence length 128, while evaluating for sequence lengths in the range $[2, 512]$. We also implement the proposed Fixed-Point framework on Mamba2 (Dao & Gu, 2024), and we compared our method to DeltaProduct (Siems et al., 2025). In Fig. 9, we plot the test accuracy for these one-layer models.

Comparing our results to DeltaProduct, we can see that the non-linearity introduced by the Fixed-Point dynamics allow for a slight improvement in the performance of the Householder products as the mixer components. Furthermore, we observe that the best performing mixer variant is still the Kroneckers model, which can successfully learn the state-tracking problem in all runs. Moreover, the FP-Mamba2 model demonsterates a better length generalization ability compared to FP-Mamba1, which we attribute to the improved underlying architecture used in the model. As shown in (Dao & Gu, 2024), Mamba2 has better recall capabilities, which can help with length generalization.

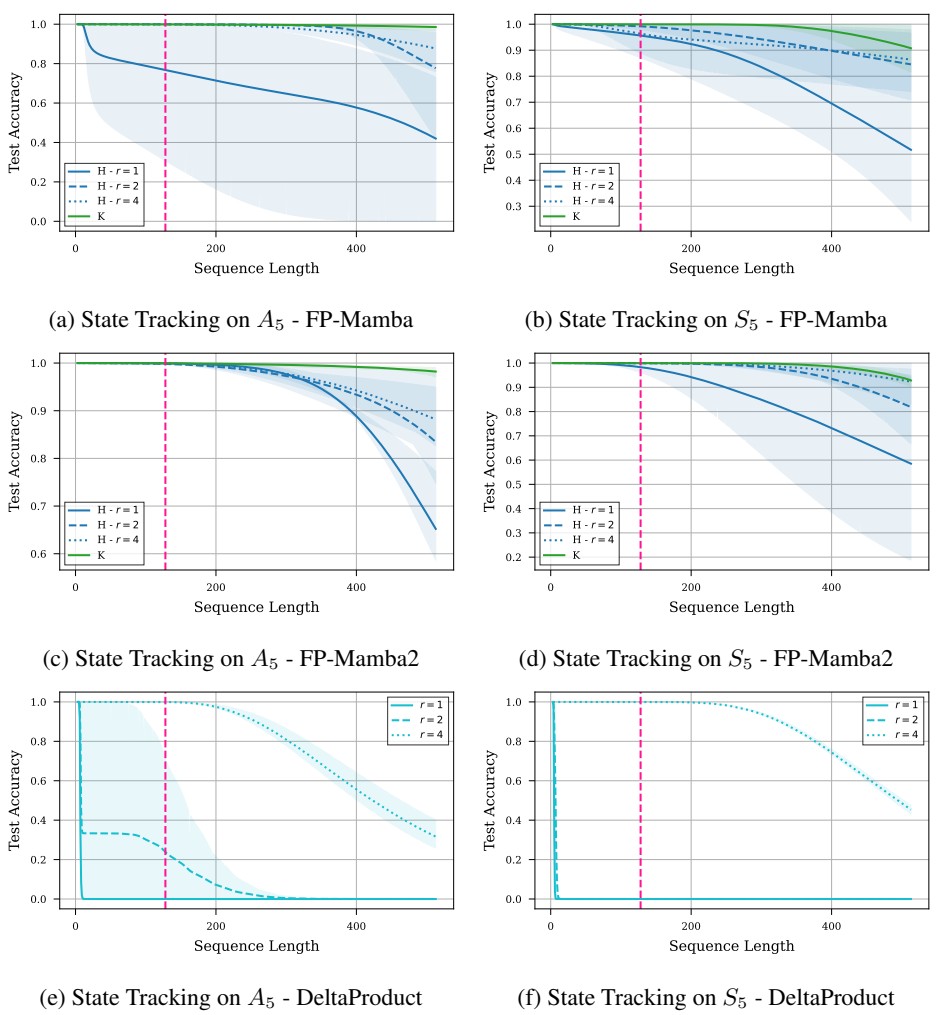

(a) State Tracking on $A_5$ - FP-Mamba

(b) State Tracking on $S_5$ - FP-Mamba

(c) State Tracking on $A_5$ - FP-Mamba2

(d) State Tracking on $S_5$ - FP-Mamba2

(e) State Tracking on $A_5$ - DeltaProduct

(f) State Tracking on $S_5$ - DeltaProduct

Figure 9: *The state-tracking experiment for train sequence length* $128$ *and evaluation sequence length* $[2, 512]$. *We omit the results of DPLR mixer due to poor performance. The figure presents **(a, b)** the results for FP-Mamba with Householder (**H**) and Kronecker (**K**) mixer, **(c, d)** the results for FP-Mamba2 with Householder (**H**) and Kronecker (**K**) mixer, and **(e, f)** the DeltaProduct method (Siems et al., 2025) for Householder mixers.*

### E.3 Reasoning on CatbAbI

In order to investigate the state-tracking ability of the fixed-point framework in a natural language setting, we perform experiments on the catbAbI dataset (Schlag et al., 2021b). catbAbI (concatenated-bAbI) is a reprocessing of the bAbI QA benchmark (Weston et al., 2015), where individual bAbI stories are stitched into one long, continuous sequence, so models must keep track of state across story boundaries. The task tries to stress-test the long-range state tracking and associative inference capabilities of sequence models beyond short, isolated contexts. Each sample in this dataset is a short story. At the end of each story, the model needs to choose a single word that is the answer to the question corresponding to the story. The responses include yes/no responses and the names of characters or locations in the story. We present the results in Table 3.

In order to observe and compare the effect of more complex mixers with the number of layers, we use $1$, $2$, and $4$ layers along with the Kronecker and Householder mixer with $r \in \{1, 2, 3\}$ reflections. Our investigation shows that increasing the number of layers seems to be reaching the point of diminishing returns very fast, while the fixed-point framework improves the performance. This observation seems to be in line with the findings of Saunshi et al. (2025), where the looped architecture seems to be providing a very helpful inductive bias for solving reasoning tasks. Comparing the performance of mixers, we observe that the Kronecker mixer under-performs compared to the Householder

| # Layers | Mamba | FP-Mamba-K | FP-Mamba-H ($r = 1$) | FP-Mamba-H ($r = 2$) | FP-Mamba-H ($r = 3$) |
|---|---|---|---|---|---|
| 1 Layer | 78.28% | 79.93% | 81.32% | 81.60% | 80.79% |
| 2 Layers | 87.08% | 84.16% | 89.08% | 87.47% | 89.55% |
| 4 Layers | 86.51% | — | — | — | — |

Table 3: *Test accuracy of the Mamba model vs. the FP-Mamba model for the Kronecker (**K**) and the Householder (**H**) channel mixers with $r \in \{1, 2, 3\}$ on the catbAbI dataset. We increase the number of layers to show the effect of having more layers on all models. The task benefits from the fixed-point dynamic, but increasing the number of layers seems to be suffering from diminishing returns.*

mixer, which we believe is in line with our observation in App. E.1, where the Kronecker mixer underperforms on tasks involving natural languages.

### E.4 Modular Arithmetic Task Results

Following Grazzi et al. (2024), we also evaluate FP-Mamba on the remaining unsolved task of the Chomsky Hierarchy of language problems introduced by Beck et al. (2024). Specifically, we focus on the mod arithmetic task with brackets. Following the setup of Grazzi et al. (2024), we train on sequence lengths 3 to 40 and report scaled accuracies on test sequences of lengths 40 to 256. For FP-Mamba, we use a 2-layer model with $r = 4$ reflections, i.e. the best performing model in the $A_5$ experiment.

In Tab. 4, we observe that a 2-layer FP-Mamba-H outperforms the baselines reported in (Grazzi et al., 2024) with a comparable number of parameters. In Fig. 10, we plot the validation accuracy as a function of the number of fixed-point iterations. We observe that the accuracy plateaus at 20 iterations, which is significantly less than the shortest and longest sequence in the validation set. Therefore, the number of iterations required by FP-Mamba-H to reach its fixed point clearly does not scale with the sequence length in this task.

| Model | Accuracy |
|---|---|
| 2L Transformer | 0.025 |
| 2L mLSTM | 0.034 |
| 2L sLSTM | **0.173** |
| 2L Mamba | 0.136 |
| 2L DeltaNet | 0.200 |
| 2L GatedDeltaProduct | **0.342** |
| 2L FP-Mamba ($r = 4$) | **0.384** |

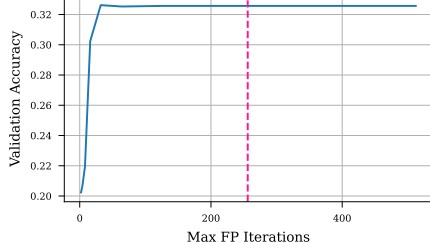

Table 4: *The accuracy of various models on modular arithmetic with brackets. We adopt the reported numbers in (Grazzi et al., 2024) evaluating baselines the extended $[-1, 1]$ eigenvalue range. Scores are commonly used scaled accuracies between $1.0$ and $0.0$ (random guessing). Highlighted is the best performance in each category.*

Figure 10: *Number of fixed-point iterations on the modular arithmetic task at test time. We report the validation accuracy after convergence for the number of fixed-point iterations caped at various values ranging from $2$ to $512$. The pink dashed line denotes the maximum sequence length during validation.*

## E.5 Effect of $\ell_{\max}$ on test performance and number of iterations $\ell^*$

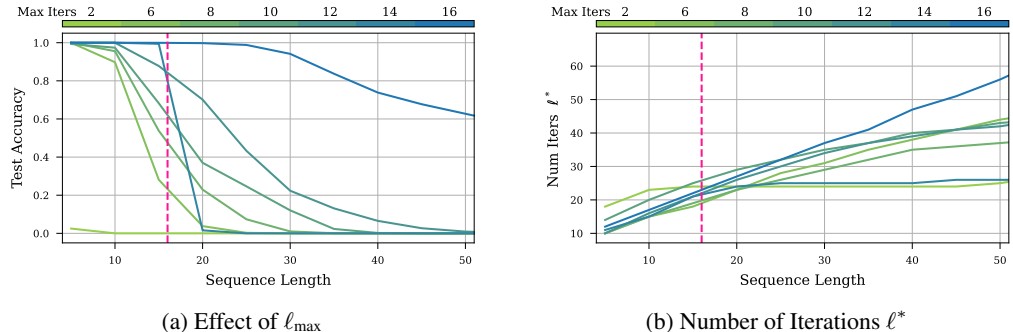

(a) Effect of $\ell_{\max}$         (b) Number of Iterations $\ell^*$

Figure 11: *The effect of $\ell_{max}$ on the performance of the model ((**a**)), and on the number of iterations $\ell^*$ ((**b**)) on the $A_5$ task. The vertical line denotes the train sequence length. All of the experiments are performed on FP-Mamba1 with a Householder mixer with $r = 1$ reflections. Results are averaged across 4 runs.*

In Fig. 11 we present the effect of the maximum number of iterations $\ell_{\max}$ during training on the accuracy and the number of iterations $\ell^*$ during inference. As we observe, the general trend is that increasing $\ell_{\max}$ improves the performance of the model. We attribute this observation to how well the model learns the task, as following Thm. 3.2, a condition for the gradients being a descent direction is for them to be computed at or close to the fixed-point. Consequently, we can see that when trained with a smaller number of iterations (small $\ell_{\max}$), the model fails to fully utilize the fixed-point by adapting $\ell^*$ to the difficulty of the task.

## E.6 Sequential vs Parallel Fixed-Point Iteration

An important detail about the fixed-point framework proposed in this paper is that it is not convex. Therefore, the fixed-point is not necessarily unique, which can be problematic in autoregressive applications because there are no guarantee that the parallel fixed-point during training will be the same as the sequential fixed-point used during inference (Schöne et al., 2025). In order to investigate this issue, we trained an FP-Mamba-H model on the $A_5$ task and compared the fixed-point computed sequentially and in parallel. We report the results in Fig. 12. We observe that the fixed-points are extremely similar, providing the possiblity of computing the fixed-point sequentially during inference.

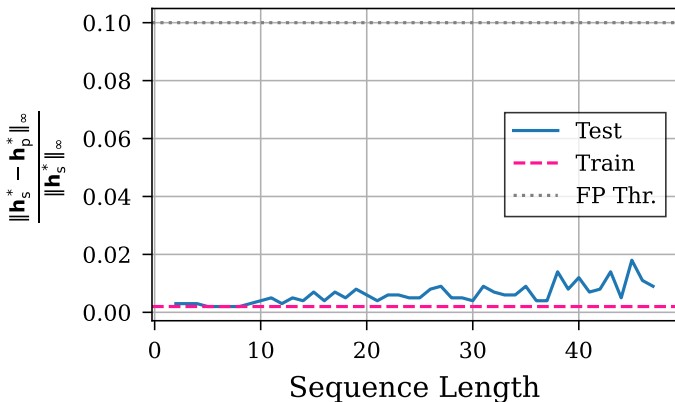

Figure 12: *The difference between the fixed-point computed sequentially (i.e., computing the fixed-point for each token separately) and the fixed-point computed in parallel (i.e., computed through Eq. 12) on the $A_5$ task trained on sequence length 16 to convergence. The x-axis denotes the test sequence length, and the y-axis the normalized difference. The dashed gray line denotes the threshold for stopping the fixed-point iterations.*

## F   Low-Rank Expressiveness

In this section, we prove that SSMs with low-rank structure can be maximally expressive under weak assumptions on the growth of the rank with hidden dimension. To do this we first place ourselves in the general setting of (Cirone et al., 2024b), accordingly we consider models given by controlled differential equations of type[3]:

$$\mathrm{d}Y_s = \sum_{i=1}^{d_\omega} A^i Y_s \mathrm{d}\omega_s^i, \quad Y_0 \in \mathbb{R}^{d_Y} \tag{19}$$

Following the notation and methodology of Cirone et al. (2024b)[B.4] ), this can be written in terms of the Signature as

$$\mathbf{Y}((A^i)_i, Y_0, \omega)_t := Y_t = \sum_{I \in \mathbb{W}_{d_\omega}} (A^I Y_0) \, S^I(\omega)_{[0,t]} \tag{20}$$

where $\mathbb{W}_{d_\omega}$ is the set of words in the alphabet $[[d_\omega]] := \{1, \ldots, d_\omega\}$ (*i.e.* $\mathbb{W}_{d_\omega} = \bigcup_{n \geq 0} [[d_\omega]]^n$ ) and for a given word $I = i_1 \ldots i_n$ with $S^I(\omega)_{[0,t]}$ we refer to the $I$th component of the *signature* tensor $S(\omega)_{[0,t]}$ *i.e.*

$$S^I(\omega)_{[0,t]} = \underbrace{\int \cdots \int}_{\substack{u_1 < \cdots < u_n \\ u_i \in [0,t]}} \mathrm{d}\omega_{u_1}^{i_1} \cdots \mathrm{d}\omega_{u_n}^{i_n}.$$

It follows directly from Eq. 20 that any linear readout of $Y_t$ can be represented as a series in signature terms. As a result, these systems are fundamentally restricted to learning functions that closely approximate these convergent series.

*Maximal expressivity* is attained when *any* finite linear combination of signature terms can be approximated by a linear readout on $Y_t$ via suitable configurations of the matrices $A^i$.

**Definition F.1.** Fix a set of paths $\mathcal{X} \subseteq C^{1-var}([0,1]; \mathbb{R}^d)$. We say that a sequence $(\mathcal{A}_N, \mathcal{Y}_N)_{N \in \mathbb{N}}$, where $\mathcal{Y}_N \subseteq \mathbb{R}^N$ and $\mathcal{A}_N \subseteq \mathbb{R}^{N \times N}$, achieves *maximal expressivity* for $\mathcal{X}$ whenever for any positive tolerance $\epsilon > 0$ and any finite linear combination coefficients $\alpha \in T(\mathbb{R}^d)$ there exist a choice of parameters $v, (A^i), Y_0$ in some $\mathbb{R}^N, \mathcal{A}_N, \mathcal{Y}_N$ in the sequence such that $v^\top \mathbf{Y}((A^i), Y_0, \omega)$. is uniformly close to $\langle \alpha, S(\omega)_{[0,\cdot]} \rangle$ up to an error of $\epsilon$ *i.e.*

$$\forall \epsilon > 0, \ \forall \alpha \in T(\mathbb{R}^d), \ \exists N \geq 0, \ \exists (v, (A^i), Y_0) \in \mathbb{R}^N \times \mathcal{A}_N^d \times \mathcal{Y}_N \text{ s.t.}$$

$$\sup_{(\omega,t) \in \mathcal{X} \times [0,1]} |\langle \alpha, S(\omega)_{[0,t]} \rangle - v^\top \mathbf{Y}((A^i), Y_0, \omega)_t| < \epsilon$$

If we are given a sequence of probabilities $\mathbb{P}_N$ on $\mathcal{A}_N^d \times \mathcal{Y}_N$ such that $\forall \epsilon > 0, \ \forall \alpha \in T(\mathbb{R}^d)$ it holds that

$$\lim_{N \to \infty} \mathbb{P}_N \left\{ \exists v \in \mathbb{R}^N \text{ s.t. } \sup_{(\omega,t) \in \mathcal{X} \times [0,1]} |\langle \alpha, S(\omega)_{[0,t]} \rangle - v^\top \mathbf{Y}((A^i), Y_0, \omega)_t| < \epsilon \right\} = 1 \tag{21}$$

then we say that $(\mathcal{A}_N, \mathcal{Y}_N, \mathbb{P}_N)_{N \in \mathbb{N}}$ achieves *maximal probabilistic expressivity* for $\mathcal{X}$.

As discussed in the main body of this work in (Cirone et al., 2024b) the authors prove that $(\mathbb{R}^{N \times N}, \mathbb{R}^N, \mathbb{P}_N)$, where $\mathbb{P}_N$ is a Gaussian measure corresponding to the classical *Glorot* initialization scheme in deep learning, achieves *maximal probabilistic expressivity* for compact sets.

Albeit expressiveness is thus maximally attained the resulting matrices $A_i$ are almost-surely dense, hence the models are not efficiently implementable. As the next result suggests, a possible alternative is given by low-rank matrices:

**Proposition F.2.** *The sequence of triplets* $(\mathbb{R}^{N \times N}, \mathbb{R}^N, \mathbb{P}_N)$ *where* $\mathbb{P}_N$ *is such that*

---

[3]For simplicity we have omitted the $\mathrm{d}\xi$ term, as the results and proof change minimally in form but not in spirit.

- *the initial value has independent standard Gaussian entries $[Y_0]_\alpha \overset{iid}{\sim} \mathcal{N}(0,1)$,*

- *the weight matrices are distributed as $A^i \overset{iid}{\sim} \frac{1}{\sqrt{Nr_N}} WM^\top$ with $W$ and $M$ independent $N \times r_N$ matrices having entries $[W]_{\alpha,\beta}, [M]_{\alpha,\beta} \overset{iid}{\sim} \mathcal{N}(0,1)$,*

- *the rank parameter $r_N$ satisfies $r_N \to \infty$ as $N \to \infty$*

*achieves* maximal probabilistic expressivity *for compact sets.*

*Proof.* Following (Cirone et al., 2024b)[B.3.5] we only need to prove a bound of type

$$\left\| \frac{1}{N}\langle A_I Y_0, A_J Y_0 \rangle_{\mathbb{R}^N} - \delta_{I,J} \right\|_{L^2(\mathbb{P}_N)} \le (\kappa(|I|+|J|))!! \; o(1) \tag{22}$$

as in the full-rank Gaussian case.

We will place ourselves in the graphical setting of (Cirone et al., 2024a) and leverage the fact that (*c.f.* (Cirone et al., 2024a)[7.1]) their results and techniques naturally hold for rectangular matrices.

In our setting $\frac{1}{N}\langle A_I Y_0, A_J Y_0 \rangle_{\mathbb{R}^N}$ corresponds to a *product graph* $G_{I,J}$ corresponding to a ladder having $2|I|+2|J|$ edges as shown in Fig. 13. We can then use (Cirone et al., 2024a)[Prop. 2] to compute the square of the $L^2$ norm in equation Eq. 22, the only difference from the dense case is that half of the vertices (excluding the "middle" one) correspond to a space of dimension $r_N$ while the rest to the standard $N$.

Since $r_N \to \infty$ and given the scaling $N^{-1}(Nr_N)^{-\frac{|I|+|J|}{2}}$, the admissible pairings of $G_{I,J}$ not of order $o(1)$ are only the leading ones. These correspond to product graphs with $\frac{|I|+|J|}{2}$ $r_N$-dimensional vertices and $\frac{|I|+|J|}{2}+1$ $N$-dimensional vertices. By the same reasoning as in the full-rank case, these are found to be just the identity pairings.

Moreover, all pairings of $G_{I,J} \sqcup G_{I,J}$ that do not result in an identity pairing in at least one of the two copies are $\mathcal{O}(\frac{1}{N \wedge r_N})$ ( instead of $\mathcal{O}(\frac{1}{N})$ ). This follows as in the full-rank case.

Since the total number of admissible pairings of $G_{I,J} \sqcup G_{I,J}$ is $(4(|I|+|J|))!!$, we conclude that equation 22 holds with $\kappa = 4$ and $o(1) := \mathcal{O}(\frac{1}{\sqrt{N \wedge r_N}})$.

$\square$

$$\frac{1}{N}\langle A_I Y_0, A_J Y_0 \rangle \quad \equiv \quad \frac{1}{N} \frac{1}{(Nr_N)^{|I|+|J|}} \quad$$

Figure 13: The *product graph* $G_{I,J}$ for $I = i_1 i_2 i_3$ and $J = j_1$.

*Remark* F.3. Following (Cirone et al., 2024a)[6.1] it's possible to prove that the $W$ and $M$ can be taken as having iid entries from a centred, symmetric but heavy tailed distribution given finiteness of even moments. This distributional choice comes useful in controlling the eigenvalues of $A = WM^\top$.

*Remark* F.4. While the proof crucially uses the assumption $r_N \to \infty$ as $N \to \infty$, at the same time we have not provided an argument against $r_N$ not diverging. In Fig. 14 we present a counterexample, showing that if $r_N$ does not diverge then the asymptotics differ from the dense ones, in particular some symmetries are "lost", impossible to recover due to unavoidable noise.

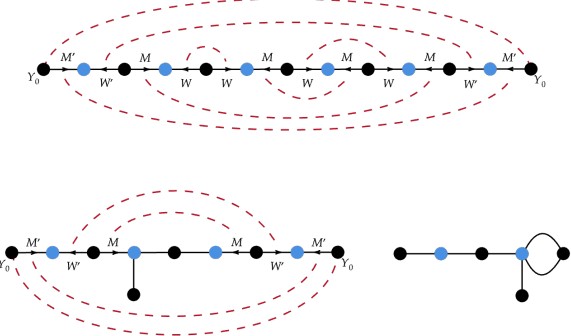

Figure 14: Admissible pairing different from the "identity" paring, but still leading to maximal asymptotic scaling in the bounded $r_N$ case. Here, $I = 12 \neq 1112 = J$, and we have highlighted in blue the vertices corresponding to the bounded dimension $r_N$. Recall that edges without arrows correspond to the matrix $\mathbf{I}$ (matrix of ones), and that two edges corresponding to matrices $A$ and $B$ which share direction and terminal vertices can be merged into the edge $A \odot B$.

