# OpenReview forum: "Fixed-Point RNNs: Interpolating from Diagonal to Dense"
_NeurIPS.cc/2025/Conference — NeurIPS 2025 spotlight_

### Official Review · Reviewer_i3G4 · 2025-06-21

**Clarity:** 3
**Significance:** 3
**Originality:** 4
**Rating:** 5
**Confidence:** 3

**Summary:**

The paper aims to bridge the gap between matrix-diagonal and matrix-dense linear RNNs by striking a balance between expressivity (which favors denser matrices) and parallelizability (a strength of diagonal RNNs that process sequences in sublinear time wrt sequence length). This is acheived by rewriting the output of dense RNN recurrance as the solution to a fix point problem, which mirrors a digagonal RNN recurrence.

Therefore, to compute the output of a dense layer under this approach, the hidden states are guessed (set to zero I believe) at first iteration, and then unrolled under the diagonal recurrence to get the next iteration (iter wrt fix point problem solver). This happens $ l^* $ times, which leaves us with time complexity of $\mathcal{O}(l^* \log(T))$, where $T$ is the sequence length.

Note that $l^* << T$, meaning that their approach is still much faster than vanilla RNNs. In the experiments, the authors empirically show superior performance, mainly in tasks where RNNs/LSTMs would excel over more modern models such as Mamba and Transformers.

**Questions:**

* Did the authors see any sort of instability during training?

* Is there a potential trade-off between the number of parameters and the required number of fixed-point iterations, i.e. $l^*$? The paper would be a lot more impactful if you show that smaller models can also trade memory, i.e. parameters, for FLOPs.

* Is there a way to reduce the numebr of fix-point interations? For example by warmstarting from the hidden states of the previous layer?

**Ethical Concerns:**

["NO or VERY MINOR ethics concerns only"]

**Final Justification:**

The contributions of the paper are novel and useful for the community, from both academic and industrial stand points. I maintained my initial rating of 5 after the discussion period.

**Limitations:**

yes, see above as well

**Paper Formatting Concerns:**

The formatting appears to be within guidelines.

**Quality:**

4

**Strengths And Weaknesses:**

Strenghts:

* The idea to generate dense RNN unrolls as fix point iterations of diagonal RNNs is novel to best of my knowledge and an idea that would benefit the research community.
* Their method is applicable to already existing SSMs and would unlock a plethora of models.
* The unrolls are very efficient. Training is also remains unaffacted of the fix point finding problem, i.e. no need to backpropagate through fix point iterations.
* The method is applied to Mamba in particular which makes the study more relavant. FP-Mamba consitently outperforms Mamab. Moreover, the authors report improved perplexity in a languge modeling task.
* They show harder tasks require more fix point iterations which proves the effectiveness of their method.
* The writing, theorems and equations are easy to follow, even though the method is somewhat complex.

Weaknesses:

* I would have appreciated a discussion regarding the effect of hidden state size in Mamba and State-Space Models (SSMs) and its relationship to expressivity. According to Koopman theory, SSMs can perfectly capture arbitrarily complex state-to-state transitions, "dynamical systems", as the state size ($d$) approaches infinity. What, then, justifies the necessity of this new approach over simply increasing $d$? Furthermore, would a larger standard Mamba model, operating under the same FLOPs, be able to outperform FP-Mamba?

* The paper would generally benefit from a more thorough benchmarking analysis of FLOPs, particularly as task complexity increases. Specifically, I would be interested in seeing how $l^∗$ scales when dealing with progressively more challenging tasks.

* The method's inability to fully capture dense RNNs, attributed to the introduction of the normalization factor $(I - \Lambda)$, warrants further discussion. Specifically, I would have liked an explanation of the precise effect this normalization has on the class of dense matrices that remain representable.

* A few minor writing errors that I was able to catch:
> - Figure 3 caption say (6) and (6).
> - I believe $h_t$ in Equation (5) should be revised to simply $h$.
> - App. C.2 under subsection 4.3 needs to be revised to C.4?
> - "DLPR" at the top of page 9, whould be revised to "DPLR."

---

> ### Author Rebuttal · Authors · 2025-07-31
>
> We thank the reviewer for their positive and detailed feedback and we’re glad that they found our paper **easy to follow**, describing a **novel idea that would benefit the research community** and could **unlock a plethora of models**. We address their suggestions as follows:
>
>
> **1. Reconcile results with Koopman theory.**
>
> Thank you for asking for clarification on this important point! Indeed, according to Koopman theory any nonlinear dynamical system can be approximated by an SSM with a large enough hidden state $d$ — as long as it is followed by an MLP tasked with computing eigenfunctions. It is important to note, however, that the form of the Koopman operator depends on the nature of the dynamical system. In the case of a scalar-valued dynamical system the state-to-state transition acts as an element-wise linear operator (like Mamba). But **the Koopman operator of a vector-valued dynamical system is generally a dense object**.  It can be diagonalized into an element-wise operator only if there exists an eigenbasis which decouples the vector-valued dynamical system. This is where the expressivity results due to Merrill et al (2024) and Cirone et al (2024) come in: **certain problems such as state-tracking cannot be expressed by a decoupled dynamical system**. In the case of state-tracking, it is easy to see that the time-varying permutation matrices which represent the linear operator cannot be co-diagonalized. We will try to make this clearer in the introduction section, and point to the Discussion in Cirone et al (2024), who directly addressed this issue mathematically in their section 4.3.
>
> Since Mamba can express progressively more coupled dynamical systems with depth, we empirically investigated scaling it’s the depth (instead of width $d$) to operate under the same runtime as FP-Mamba. We hope that answer we gave to the other reviewers addresses your question:
>
> **2. Make compute trade-offs more transparent.**
>
> We agree that the main submission provided insufficient guidance on the compute trade-offs of our FP-RNN approach when restricting the number of iterations. Therefore, we had already included plot to relate compute time to generalisation on the state-tracking task $A\_5$ in Appendix E.4, Figure 11 and now provide an analysis here:
>
> *”Wall clock time is plotted against the longest test sequence length with > 90% accuracy for every model. While baselines of increasing depth $\\in \\{1, 2, 4, 6, 8\\}$ cannot generalize beyond the training sequence length 16 (horizontal pink line), our proposed framework allows to **achieve much higher generalization by scaling training time** through the number of fixed-point iterations $\\ell\_\\text{max} \\in \\{2, 4, 8, 16\\}$. There is **further room to improve efficiency**, as suggested by a simple randomization scheme (gray stars) where $\\ell\_{\\text{max}} \\sim\\Gamma(4,1)$ is sampled from a Gamma distribution with mean $4$ for every batch.”*
>
> We will include the above analysis in Sec 4.5 and move Figure 11 to the main text (replacing Figure 7).
>
> **3. Scaling of $\\ell^*$ with progressively more challenging tasks**
>
> We would love to provide benchmarking analysis, but unfortunately the NeurIPS rebuttal format does not allow for rich media. Therefore, we will resort to explaining the theoretical cost of executing a single layer of our method vs. the underlying sequence mixer used (Mamba) instead. Denoting the sequence length by $T$, the state size by $N$, and the number of iterations to convergence by $\\ell^*$, the FLOPS are:
>
> | Model | Sequence Mixing Cost | Channel Mixing Cost |
> | --------      | -------     | -------   |
> | Mamba | $\\mathcal{O}(TN)$ (forward), $\\mathcal{O}(TN)$ (backward) |$0$ |
> | FPMamba (H, DPLR) | $\\mathcal{O}(TN \\ell^*)$ (forward), $\\mathcal{O}(TN)$ (backward) | $\\mathcal{O}(T N\\ell^*)$ (forward), $\\mathcal{O}(TN)$ (backward) |
> | FPMamba (K) | $\mathcal{O}(TN \\ell^*)$ (forward), $\\mathcal{O}(TN)$ (backward) | $\\mathcal{O}(T N \\sqrt{N} \\ell^{*})$ (forward), $\\mathcal{O}(TN\\sqrt{N})$ (backward)|
>
>
> Note that the overall cost of FP-Mamba is *$\\ell^\*$ times the cost of channel+sequence mixing* for the forward pass while **no additional cost for the iteration occurs for the backward pass**. While the sequence mixing cost is inherited from Mamba, the **channel mixing incurs an additional cost that is sub-quadratic in state size and linear in sequence length** in all three variants of the mixers investigated in this paper.
>
> Figure 3c depicts the value of $\\ell^\*$ in a progressively more challenging task. In this state tracking task $A\_5$, the difficulty corresponds to the number of state transitions, which is equal to the sequence length $T$. From the figure, we observe a linear relationship between $\\ell^\*$ and $T$, indicating that, at least in the state tracking problem, $\\ell^\*$ scales linearly with the difficulty of the task. Note, however, that on simpler tasks (Figures 5b, 7, 10) the required number of fixed-point iterations $\\ell^\*$ remains well below the sequence length $T$. Finally, the **overall FLOPs are scaled by the task-specific** $\\ell^\*$ according to the cost in the table above. We will update the paper with this analysis.
>
>
>
>
> **4. Effect of normalization factor $\\mathbf{(I-\\Lambda\_t)}$ on class of matrices $A\_t$.**
>
> Thank you for pointing out the missing description of the representable dense matrices $\\mathbf{A}\_t$ in the presence of the normalization factor $(\\mathbf{I}-\\mathbf{\\Lambda}\_t)$. To drive $\\mathbf{A}\_t$, let us start by assuming a fixed-point was found according to Thm 3.1:
>
> $$h\_t^* = \\mathbf{\\Lambda}\_t h\_{t-1}^* + (\\mathbf{I}-\\mathbf{\\Lambda}\_t) (\\mathbf{Q}\_t\\mathbf{B}\_t x\_t + (\\mathbf{I} -\\mathbf{Q}\_t)h\_t^*)$$
>
> Rearranging the terms allows to move $h\_t^*$ to the other side:
>
> $$(\\mathbf{I} - (\\mathbf{I} - \\mathbf{\\Lambda}\_t)(\\mathbf{I} - \\mathbf{Q}\_t))h\_t^* = \\mathbf{\\Lambda}\_t h\_{t-1}^* + (\\mathbf{I}-\\mathbf{\\Lambda}\_t) \\mathbf{Q}\_t\\mathbf{B}\_t x\_t$$
>
> Therefore, $\\mathbf{A}\_t = (\\mathbf{I} - (\\mathbf{I} - \\mathbf{\\Lambda}\_t)(\\mathbf{I} - \\mathbf{Q}\_t))^{-1}  \\mathbf{\\Lambda}\_t = (\\mathbf{\\Lambda}\_t^{-1}(\\mathbf{I} - (\\mathbf{I} - \\mathbf{\\Lambda}\_t)(\\mathbf{I} - \\mathbf{Q}\_t)))^{-1} = (\\mathbf{\\Lambda}\_t^{-1} - (\\mathbf{\\Lambda}\_t^{-1} - \\mathbf{I})(\\mathbf{I} - \\mathbf{Q}\_t))^{-1} =(\\mathbf{I} + (\\mathbf{\\Lambda}\_t^{-1} - \\mathbf{I})\\mathbf{Q}\_t)^{-1}$
>
> Following the standard assumptions that $0\\preceq \\mathbf{\\Lambda}\_t, \\mathbf{Q}\_t\\preceq \\mathbf{I}$, the matrix $\\left(\\mathbf{I}+\\left(\\mathbf{\\Lambda}\_t^{-1} - \\mathbf{I}\\right)\\mathbf{Q}\_t\\right)$ is full rank and $\\succeq \mathbf{I}$. **Therefore its inverse $\\mathbf{A}\_t$ exists and is contractive.** The expressivity of $\\mathbf{A}\_t$ is only limited if  $\\mathbf{\\Lambda}\_t\\approx \\mathbf{I}$. This however, would also be problematic for a diagonal SSM and therefore the Mamba initialization is biased towards $\\mathbf{\\Lambda}\_t \\prec \\mathbf{I}$. Thus, we argue that in practice, these modifications do not pose a significant problem. We will make sure to refer to this analysis when we introduce normalization in Sec. 3.1.
>
>
>
> **Questions:**
>
> 1. *Are FP-RNNs unstable during training?* Although a large portion of our theoretical contributions focus on this issue, we notice that under certain conditions we can still face instability in training. Specifically, we notice that for long sequences in the state tracking tasks ($A\_5$, $S\_5$), where the difficulty of the task scales linearly with the sequence length, training can become unstable. In this case, we notice that incorporating off-the-shelf stability techniques, such as weight normalization, can significantly help with this issue (Gent et al, 2023).
> 2. *Are there trade-offs between the number of parameters and the required number of fixed-point iterations?* In our experiments, we indeed observe a relationship between the parameter numbers and the required number of fixed-point iterations for finding the fixed point. However, this only happens when the model is modified in depth, and not in width. Specifically, we observe that when increasing the depth of the model for more challenging tasks, the number of iterations indeed reduces.
> 3. *Is there a way to reduce the number of fixed-point iterations?* We have several potential solutions to reducing the number of fixed-point iterations. Firstly, we observe that off-the-shelf optimization techniques do not perform as well as fixed-point iterations near the initial point of the hidden state, but are much faster to find the exact solution once the hidden state is in the vicinity of the fixed-point. Consequently, we believe a hybrid approach would be possible to significantly convergence. We also introduce some techniques in Appendix B.4 for this purpose. Finally, as we noted in our response in the previous question, we observe that for deeper networks, the fixed-point becomes much easier to find. This may indicate the existence of a structure in the mixer (e.g. positive semi-definiteness) that can significantly improve the cost of finding the fixed-point.
>
>
> We thank the reviewer for the list of minor issues which we will fix in the camera ready version. We hope that our response addresses the reviewer’s concerns and would be interested to hear if they are considering raising their score to help increase the visibility of our work during the conference. In case there are any remaining questions, we are of course happy to resolve them during the discussion period.
>
>
> ---
>
>
> Merrill et al (2024), *”The Illusion of State in State-Space Models”*, https://arxiv.org/abs/2404.08819
>
> Cirone et al. (2024), *”Theoretical Foundations of Deep Selective State-Space Models”*, https://arxiv.org/abs/2402.19047
>
> Geng et al. (2023). *"Torchdeq: A library for deep equilibrium models."* https://arxiv.org/abs/2310.18605

---

> > ### Comment · Reviewer_i3G4 · 2025-08-04
> >
> > I thank the authors for their thorough response. They have addressed my previous questions and concerns. Therefore, I will maintain my score of 5.

---

### Official Review · Reviewer_DU4S · 2025-06-26

**Clarity:** 3
**Significance:** 4
**Originality:** 3
**Rating:** 6
**Confidence:** 4

**Summary:**

The authors propose a new parameterization for (implicit) dense linear RNNs as fixed-points of diagonal linear RNNs. The resulting sequence mixer shows superior expressivity which is shown both theoretically and empirically on state tracking tasks. The authors clearly explain how their proposed model allows for a tunable trade-off between expressivity and efficiency (i.e., parallelizablity) by varying the number of iterations. Their experiments show interesting results on adaptive computation that are well-aligned with their theory. Their model performs well on synthetic toy tasks as well as a small language modeling experiment.

**Questions:**

Could you elaborate on how, once the recurrence depends on the hidden state (as in Section 3.4), the theoretical guarantee of a unique fixed point from Theorem 3.1 remains? Since this state-dependence breaks the uniform contraction assumption of Theorem 3.1 and can admit multiple solutions $h^*$, under what additional conditions (if any) could you still prove convergence—-or at least uniqueness—-of the fixed-point iteration in this richer setting? My current understanding is that the empirical iteration condition is not guaranteed to hold and the system may iterate forever. It would be interesting to know under what conditions this might happen, so I would be curious if you have any results or ideas on this.

Can you explain why the language modeling experiments in Appendix E.1 use the Householder mixer, while the experiments in the main paper showed the Kronecker to be most performant? Is there any guidance on which structure performs best on different tasks?

**Ethical Concerns:**

["NO or VERY MINOR ethics concerns only"]

**Final Justification:**

The authors addressed all my concerns and incorporated all suggestions.

**Limitations:**

See comment about the authors' claims of state-of-the-art result.

**Quality:**

3

**Strengths And Weaknesses:**

**Strengths**
- theoretically grounded, novel, and clean method for smoothly interpolating between diagonal linear RNNs and dense RNNs. Their approach cleanly separates time-recurrence (diagonal gate) from channel mixing, simplifying analysis and gradient flow. Shows how a single RNN step can be "deepened" via fixed-point iteration without stacking layers.
- Demonstrates competitive or improved accuracy on synthetic copying tasks, and shows some results on language modeling as well.
- very well-written paper with didactic explanations, it was a pleasure to read.
- clearly points to new research directions in the field and opens new opportunities.

**Weaknesses**
- Time complexity might be $O(T^2)$ which would be worse than nonlinear RNNs, and the analysis in the paper is not convincing that this is *not* the case. Indeed, Figure 3c leads me to believe that the number of iterations indeed does scale with sequence length, leading to quadratic complexity.
- Not clear how well the interpolation works when restricting the number of fixed point iterations - Figure 1 is the only result shown in the paper but I would be curious about the comparison to other linear sequence mixers. Does the proposed FP-RNN with maximum one iteration perform as well as Mamba? Little guidance on these trade-offs is offered in the paper.
- Limited ablations: there is no detailed study of how iteration count, tolerance, or mixer depth affect performance.
- Limited results on real-world benchmarks like language modeling (only compared to Mamba-1, not other models). It seems that their proposed FP-RNN outperforms Mamba-1 on language modeling but scalability to real world problems is not demonstrated very convincingly.
- The claim of state-of-the-art performance in the abstract is not clearly backed up in the main text. It is advisable for the authors to clarify their exact new state-of-the-art results in the results section in more detail, and to qualify their claim in the abstract.

Minor:
- repeated sentence lines 42-43 and 44-45
- footnote 1 grammar: "allows to increasing"
- first introduction of the name FPMamba is a bit unexpected, might be better for the reader if it's introduced directly in the abstract and introduction as a contribution of the paper
- reference needs brackets in line 134
- broken sentence in line 140-141, 156
- the task descriptions is not referenced in the main text (appendix D)
- figure 7 is not referenced in the text

---

> ### Author Rebuttal · Authors · 2025-07-31
>
> We thank the reviewer for their positive and detailed feedback and we’re glad that they found our paper **a pleasure to read**, describing a **theoretically grounded, novel, and clean method** that **clearly points to new research directions**. We address their suggestions as follows:
>
> **1. Make compute trade-offs more transparent.**
>
> We agree that the main submission provided insufficient guidance on the compute trade-offs of our FP-RNN approach when restricting the number of iterations. Therefore, we had already included plot to relate compute time to generalisation on the state-tracking task $A\_5$ in Appendix E.4, Figure 11 and now provide an analysis here:
>
> *”Wall clock time is plotted against the longest test sequence length with > 90% accuracy for every model. While baselines of increasing depth $\\in \\{1, 2, 4, 6, 8\\}$ cannot generalize beyond the training sequence length 16 (horizontal pink line), our proposed framework allows to **achieve much higher generalization by scaling training time** through the number of fixed-point iterations $\\ell\_\\text{max} \\in \\{2, 4, 8, 16\\}$. There is **further room to improve efficiency**, as suggested by a simple randomization scheme (gray stars) where $\\ell\_{\\text{max}} \\sim\\Gamma(4,1)$ is sampled from a Gamma distribution with mean $4$ for every batch.”*
>
> We will include the above analysis in Sec 4.5 and move Figure 11 to the main text (replacing Figure 7). Thank you for this excellent suggestion which makes the paper stronger!
>
>
> **2. More evaluations on real-world benchmarks.**
>
> Thank you for sharing this perspective! It’s true that the main goal of the paper is to introduce a framework which solves the theoretically motivated state-tracking task and we therefore focus on this aspect in the evaluation. Nevertheless, we evaluate a small ~160M FP-Mamba language model trained on 2B tokens of Fine-Web (Penedo et al, 2024) in Appendix E.1. In addition, we now provide **new results of FP-Mamba on the reasoning benchmark CatbAbi** (Schlag et al, 2021):
>
> | Accuracy | Mamba | FP-Mamba (K) | FP-Mamba (H1) | FP-Mamba (H2) | FP-Mamba (H3) |
> | --------      | -------     | -------   | -------   | -------   | -------   |
> | 1 Layer    | 78.28 %  | 79.93 % | 81.32 % | 81.60 % | 80.79 % |
> | 2 Layers  | 87.08 %  | 84.16 % | 89.08 % | 87.47 % | 89.55 % |
> | 4 Layers  | 86.51 %  | - % | - % | - % | - % |
>
> In this experiment, we report the test accuracy, using the settings of (Schlag et al, 2021). We optimize the learning rates on Mamba, and use the same learning rate to train FP-Mamba. The vanilla Mamba model hits at the diminishing returns point with more layers, while our proposed dense variant further improves the accuracy with more dense layers. Furthermore, we can see that in this case, the Householder mixer performs better than the Kronecker mixer, indicating that the Householder mixer is more suitable for text data. We observed a similar pattern in language modeling, where FPMamba achieves a perplexity of 26.00 with the Kronecker mixer on the test set, while with a single Householder component it achieves the perplexity of 25.24. The vanilla Mamba achieves a perplexity of 27.01 in this case.
>
> We agree with the reviewer that these initial results on real-world tasks are indeed an important signal to the community and we will thus place them more prominently in the paper. We believe that **our old language and new reasoning results point into a promising direction** and we would like to include more benchmarks in the camera-ready version if the reviewer has some concrete suggestions. However, we believe that a thorough, large-scale study to investigate which application domains benefit from our verified state-tracking capabilities is beyond the scope of this work.
>
> **3. Further discuss required number of iterations.**
>
> Yes, we do observe a complexity of $O(T^2)$ on state-tracking in Figure 3c. However, as we discuss in L365-365, we believe that *“this is not necessarily a disadvantage if the model is capable of adapting its sequential steps to the difficulty of the task, with negligible cost for the least demanding tasks*”. Indeed, on copying (Figure 5b), language modelling (Figure 6), and modular arithmetic (Figure 10) we observe the required number of fixed-point iterations $\ell^*$ is well below the sequence length $T$ suggesting that **the model adapts to $O(T)$ complexity on simpler tasks**. Finally, a new theoretical study suggests that logarithmic depth might be both necessary and sufficient to achieve state tracking from an expressivity perspective (Merrill et al 2025). In future work, we will investigate how the number of fixed-point iterations could be reduced to this theoretical lower bound in practice. We will make sure to discuss this better in Sec. 4.5 of the paper, thank you for raising this concern!
>
>
> **4. Ablate iteration count, tolerance, and mixer depth.**
> Thank you for the suggestion. We first note that the iteration count is not controllable by us, since it is the model that determines when it reaches the fixed-point. So we assume you are referring to the maximum number of fixed-point iterations. We agree these ablation studies are valuable, and **the results in Figure 1 provide a preliminary analysis of these parameters**. Our work ultimately focuses on ablating core architectural components, such as the choice of mixer, as we view iteration count and tolerance as solver-specific hyperparameters. But we are happy to expand on these initial findings with a more detailed study for the camera-ready version if needed.
>
> Regarding the mixer-depth choice, we would like to kindly ask you to further elaborate on what you are exactly interested in. In case your question is with regards to the number of Householder (H) components, **we have investigate the effect of more Householder components in the paper (up to 4)**.
>
> **5. Highlight state-of-the-art results in the paper.**
> Thank you for your attention to detail. **Our claim on state-of-the-art results in the paper specifically refer to the A5/S5 task**, for which we believe our reported result for figure 5-a and 5-b - achieving near perfect $>\\times 3$ length generalization - is the best performance reported for these tasks for training in sequence length 16. Furthermore, using the setting proposed for the DeltaProduct method (Siems et al, 2025) for training in sequence length 128, we achieve better $\\times 4$ length generalization than their proposed method on both the $A\_5$ and $S\_5$ tasks (average of 3 separate runs), the result of which are available in the following table:
> | Accuracy | GatedDeltaProduct (H4) | FP-Mamba1 (K) | FP-Mamba1 (H4) | FP-Mamba2 (K) | FP-Mamba2 (H) |
> | --------      | -------     | -------   | -------   | -------  | -------   |
> | A5    | 23.90 %  | 99.55 % | 87.68 % | 98.21 % | 88.19 % |
> | S5  | 45.42 %  | 90.75 % | 86.37 % | 92.82 % | 92.20 % |
>
> We will make the distinction in the abstract of the paper about this point.
>
> **Questions:**
>
> 1. *What are theoretical guarantees once the recurrence depends on the hidden state?* While we did not establish a formal proof, we’re happy to share some ideas on how to prove convergence in this richer setting. To start, we agree that a state-dependent iteration can admit multiple solutions. However, note that in the time dimension, sequence mixing occurs only uni-directionally. This allows for an inductive argument in both dimensions: Since $Q\_0^{\\ell}$ is contractive, depends only on the input and no (varying) hidden state, the iteration in depth on the first time step $h\_0^{\\ell}$ converges (similar to Thm 3.1). Once $h\_0^{\\ell}$ converged and does not change anymore, $Q\_1^{\\ell}$ will not change anymore and allow for $h\_1^{\\ell}$ to converge. This way, the iterations $h\_t^{\\ell}$ converge sequentially in time and the system cannot iterate forever.
>
> 2. *Is there any guidance on which structure performs best on different tasks?* Since the submission, we have ablated the mixers on all tasks and conclude that the Kronecker structure seems to be a good default. The language experiment, however, is the only one where Householder mixers perform slightly better than Kronecker mixers. We will add these new results to the camera-ready version of the paper.
>
>
> We thank the reviewer for the list of minor issues which we will fix in the camera ready version. We hope that our response addresses the reviewer’s concerns and would be interested to hear if they are considering raising their score to help increase the visibility of our work during the conference. In case there are any remaining questions, we are of course happy to resolve them during the discussion period.
>
>
>
>
> ---
>
> Penedo et al. (2024), *"The FineWeb Datasets: Decanting the Web for the Finest Text Data at Scale"*, https://arxiv.org/abs/2406.17557
>
> Schlag et al. (2021), *"Learning associative inference using fast weight memory."*, https://arxiv.org/abs/2011.07831
>
> Merrill et al. (2025), "A Little Depth Goes a Long Way: The Expressive Power of Log-Depth Transformers.", https://arxiv.org/abs/2503.03961
>
> Siems et al. (2025), "Deltaproduct: Improving state-tracking in linear rnns via Householder products." https://arxiv.org/abs/2502.10297

---

> > ### Comment · Reviewer_DU4S · 2025-08-05
> >
> > I thank the reviewers for their detailed response. I appreciate the clarifications for the paper presented in points 1, 3, 5 and question 1, as well as the additional results presented in point 2 and question 2.
> >
> > My main concern regarding ablations remains; the paper does not present a systematic study of the solver hyperparameters that govern the compute-accuracy trade-off - without this, it's hard to isolate the effect of the different parameters. I would have liked to see a study on how the maximum iteration budget affets performance on the three tested tasks with the distribution of actual iterations per layer/token reported so that the reader can judge the real sequential cost. Similarly, an ablation on the convergence threshold/tolerance and mixer-depth would be helpful. Figure 1 shows $r \in \{ 1, 2, 4 \}$ and it would be interesting to show if further increases of $r$ lead to further increases in performance, or if performance increases diminish for $r>4$. I will keep my score of 5 and wish the authors all the best with their paper and I look forward to seeing their follow-up research.

---

> ### Author Response · Authors · 2025-08-09
>
> Thank you for taking the time to explain in so much detail the experimental setup you would have liked to see. This is very valuable feedback since there are many possible configurations for ablations and we believed that we had covered the most important aspects. We particularly liked your suggestion to relate **maximum iteration budget** $\\ell\_{\\text{max}}$ at *training time* to the effective number of iterations $\\ell^*$ at *test time* and ran the corresponding experiment at least for $A\_5$ in the past days. We will create a new ablation section in the Appendix and add results depicted in the table below as a plot. We plan to add similar plots to elucidate the **effect of the tolerance** $\epsilon$ at *test-time* for models with tolerance $\epsilon=0.1$ at *training-time*. Although we might not see your response until the final reviews are released, we would still be interested to hear if this is the ablation you meant or what you would deem more meaningful.
>
> Finally, we want to apologize for the confusion caused by the legend in Fig.1, where $r\\in\{1,2,4\}$ denotes the **number of reflections r of the Householder (H)** sequence mixer $Q\_t$. We updated all the legends now to make this clearer. We believe that investigating the effect of different mixer structures is a crucial result of our work beyond just ablations. Therefore we had added more results on all the tasks, including **rank r of the Diagonal Plus Low-Rank (DPLR)** and the **hyperparameter-free Kronecker (K)** sequence mixer $Q\_t$ after the submission. For the Householder, we observed in preliminary experiments that increasing the number of reflections to $r>4$ yields diminishing returns especially since increasing the rank also increases the computational cost. This is again visible in Fig. 11 of Appendix E.4, where the best performing model ($\\ell_{\\text{max}}=16$ at $r=4$) is 25% slower than the next best model ($\\ell_{\\text{max}}=16$ at $r=2$) but generalizes only to 10% longer sequences.
>
> ## Ablation train-time $\\ell\_{\\text{max}}$ vs performance and test-time $\\ell^*$
>
> | seqlen | l_max=2 | l_max=6 | l_max=8 | l_max=10 | l_max=12 | l_max=14 | l_max=16 |
> | --- | --- | --- | --- | --- | --- | --- | --- |
> | 5	| 2.5% @18l*	 | 99.99% @10l*	 | 100.0% @10l*	 | 99.34% @14l*	 | 100.0% @10l*	 | 100.0% @11l*	 | 100.0% @12l*	 |
> | 10	| 0.0% @23l*	 | 89.76% @15l*	 | 95.54% @15l*	 | 97.38% @20l*	 | 99.99% @16l*	 | 100.0% @15l*	 | 99.98% @17l*	 |
> | 15	| 0.0% @24l*	 | 28.09% @18l*	 | 53.8% @19l*	 | 68.25% @25l*	 | 87.78% @21l*	 | 99.41% @21l*	 | 99.94% @22l*	 |
> | 20	| 0.0% @24l*	 | 3.8% @23l*	 | 22.95% @23l*	 | 37.01% @29l*	 | 70.14% @26l*	 | 1.57% @24l*	 | 99.77% @27l*	 |
> | 25	| 0.0% @24l*	 | 0.28% @28l*	 | 7.37% @26l*	 | 24.65% @32l*	 | 43.3% @30l*	 | 0.0% @25l*	 | 98.83% @32l*	 |
> | 30	| 0.0% @24l*	 | 0.01% @31l*	 | 1.0% @29l*	 | 12.05% @35l*	 | 22.36% @34l*	 | 0.0% @25l*	 | 94.15% @37l*	 |
> | 35	| 0.0% @24l*	 | 0.0% @35l*	 | 0.07% @32l*	 | 2.31% @37l*	 | 13.1% @37l*	 | 0.0% @25l*	 | 83.71% @41l*	 |
> | 40	| 0.0% @24l*	 | 0.0% @38l*	 | 0.0% @35l*	 | 0.21% @40l*	 | 6.55% @39l*	 | 0.0% @25l*	 | 73.85% @47l*	 |
> | 45	| 0.0% @24l*	 | 0.0% @41l*	 | 0.0% @36l*	 | 0.02% @41l*	 | 2.69% @41l*	 | 0.0% @26l*	 | 67.74% @51l*	 |
> | 50	| 0.0% @25l*	 | 0.0% @44l*	 | 0.0% @37l*	 | 0.0% @43l*	 | 0.92% @42l*	 | 0.0% @26l*	 | 62.72% @56l*	 |
> | 55	| 0.0% @27l*	 | 0.0% @46l*	 | 0.0% @38l*	 | 0.0% @44l*	 | 0.27% @44l*	 | 0.0% @26l*	 | 57.77% @62l*	 |
> | 60	| 0.0% @31l*	 | 0.0% @47l*	 | 0.0% @38l*	 | 0.0% @46l*	 | 0.07% @45l*	 | 0.0% @26l*	 | 53.15% @67l*	 |
>
>
> Effect of train-time iteration budget on performance and effective test-time iterations for a 1-layer FP-Mamba with $r=1$ Householder reflections on the state-tracking task $A\_5$. Each cell contains the test accuracy in % and the corresponding test-time number of iterations $l^*$. Reported are averages over 4 runs.

---

### Official Review · Reviewer_TEck · 2025-07-03

**Clarity:** 3
**Significance:** 2
**Originality:** 3
**Rating:** 5
**Confidence:** 3

**Summary:**

The paper introduces Fixed-Point RNNs (FP-RNNs), a novel framework that models dense linear recurrent neural networks as the fixed points of diagonal RNNs. This approach enables a continuous trade-off between computational efficiency and expressivity by varying the number of fixed-point iterations, effectively interpolating from parallelizable diagonal structures to expressive dense ones. The method ensures stability through careful parameterization and uses structured mixers (e.g., Kronecker products, Householder reflections) to enable efficient and scalable computation.

The authors instantiate this framework in FP-Mamba, an extension of the Mamba selective state-space model, demonstrating state-of-the-art performance on tasks that require memory and state-tracking, such as the copy task and sequence length generalization benchmarks (A5/S5). Notably, FP-Mamba adapts the depth of computation dynamically based on task complexity and outperforms traditional RNNs and SSMs in expressivity, while maintaining better parallelism than classical RNNs. The work opens a path toward more expressive and efficient sequence models that unify the strengths of recurrent and state-space architectures.

**Questions:**

1. In practice, are there times or examples where convergence of the FP-RNN would be very slow? Any intuition about when or why?
2. How should one select one of the mixer structures in practice?
3. Could you provide concrete cost measurements (FLOPs, memory) comparing FP-RNNs to existing models?

**Ethical Concerns:**

["NO or VERY MINOR ethics concerns only"]

**Final Justification:**

The authors addressed my main concerns, showing performance on additional tasks and discussing compute time trade-offs in the main text. I have updated my score accordingly.

**Limitations:**

yes

**Quality:**

3

**Strengths And Weaknesses:**

Strengths
- The paper is well written and easy to follow. The intro and background were especially clear.
- The fixed-point RNN framework is an elegant way to flexibly interpolate between diagonal and dense RNNs, enabling adaptive computation.
- The paper carefully parameterizes the RNN in order to ensure stable dynamics.

Weaknesses:
- Limited empirical scope: the experiments focus on synthetic benchmarks, with little evaluation on real-world tasks.
- A quantitative comparison table of compute time, memory usage, and iteration cost vs baselines would make the trade-offs more transparent.
- The paper lacks practical intuition or heuristics for choosing the number of fixed point iterations

---

> ### Author Rebuttal · Authors · 2025-07-31
>
> We thank the reviewer for their valuable feedback and we appreciate their assessment of our paper as **well written** and **easy to follow**, describing a **novel, elegant method** that **opens a path** towards improved sequence models. We address their suggestions as follows:
>
> **1. Make compute trade-offs more transparent.**
>
> We agree that the main submission provided insufficient guidance on the compute trade-offs of our FP-RNN approach  when restricting the number of iterations. Therefore, we had already included plot to relate compute time to generalisation on the state-tracking task $A\_5$ in Appendix E.4, Figure 11 and now provide an analysis here:
>
> *”Wall clock time is plotted against the longest test sequence length with > 90% accuracy for every model. While baselines of increasing depth $\\in \\{1, 2, 4, 6, 8\\}$ cannot generalize beyond the training sequence length 16 (horizontal pink line), our proposed framework allows to **achieve much higher generalization by scaling training time** through the number of fixed-point iterations $\\ell\_\\text{max} \\in \\{2, 4, 8, 16\\}$. There is **further room to improve efficiency**, as suggested by a simple randomization scheme (gray stars) where $\\ell\_{\\text{max}} \\sim\\Gamma(4,1)$ is sampled from a Gamma distribution with mean $4$ for every batch.”*
>
> We will include the above analysis in Sec 4.5 and move Figure 11 to the main text (replacing Figure 7). Thank you for this excellent suggestion which makes the paper stronger!
>
>
> **2. More evaluations on real-world benchmarks.**
>
> Thank you for sharing this perspective! It’s true that the main goal of the paper is to introduce a framework which solves the theoretically motivated state-tracking task and we therefore focus on this aspect in the evaluation. Nevertheless, we evaluate a small ~160M FP-Mamba language model trained on 2B tokens of Fine-Web (Penedo et al, 2024) in Appendix E.1. In addition, we now provide **new results of FP-Mamba on the reasoning benchmark CatbAbi**  (Schlag et al, 2021):
>
> | Accuracy | Mamba | FP-Mamba (K) | FP-Mamba (H1) | FP-Mamba (H2) | FP-Mamba (H3) |
> | --------      | -------     | -------   | -------   | -------   | -------   |
> | 1 Layer    | 78.28 %  | 79.93 % | 81.32 % | 81.60 % | 80.79 % |
> | 2 Layers  | 87.08 %  | 84.16 % | 89.08 % | 87.47 % | 89.55 % |
> | 4 Layers  | 86.51 %  | - | - | - | - |
>
> In this experiment, we report the test accuracy, using the settings of (Schlag et al, 2021). We optimize the learning rates on Mamba, and use the same learning rate to train FP-Mamba. The vanilla Mamba model hits the diminishing returns point with more layers, while our proposed dense variant further improves the accuracy with more dense layers. Furthermore, we can see that in this case, the Householder mixer performs better than the Kronecker mixer, indicating that the Householder mixer is more suitable for text data. We observed a similar pattern in language modeling, where FPMamba achieves a perplexity of 26.00 with the Kronecker mixer on the test set, while with a single Householder component it achieves the perplexity of 25.24. The vanilla Mamba achieves a perplexity of 27.01 in this case.
>
> We agree with the reviewer that these initial results on real-world tasks are indeed an important signal to the community and we will thus place them more prominently in the paper. We believe that **our old language and new reasoning results point into a promising direction** and we would like to include more benchmarks in the camera-ready version if the reviewer has some concrete suggestions. However, we believe that a thorough, large-scale study to investigate which application domains benefit from our verified state-tracking capabilities is beyond the scope of this work.
>
> **3. Provide intuition/heuristics for choosing the number of fixed-point iterations.**
>
> As described in L222 and L298, the fixed-iteration stops once the relative distance between iterates falls below a threshold of 0.1, so **the number of fixed-point iterations is chosen by the model itself and not the user**. In order to avoid excessive computation on outlier samples during training, we usually set the maximum number of iterations $\\ell\_{max}$ to the sequence length $T$ and apply other heuristics as described in Appendix B.4.
>
>
> **Questions:**
>
> 1. *Are there examples where the convergence of the FP-RNN is very slow?* We observe different convergence speeds depending on the application. For example state-tracking tasks are very hard to solve, whereas language modelling is easier and converges usually in a few iterations as depicted in Figure 6.
>
> 2. *How should the mixer structure be selected in practice?* We suggest empirically evaluating the best performing mixer structure depending on the specific application. In the meantime, we have ablated the mixers on all tasks and conclude that the Kronecker structure seems to be a good default for state tracking, and the Householder structure more suitable for language and copying. However, we encourage the community to investigate more channel mixer structures under the FP-RNN framework.
>
> 3. *Could you provide concrete cost measurements comparing FP-RNNs to existing models?* As discussed before, we compare the runtime of FP-RNN to existing models in Appendix E.4, Figure 11 and will move this discussion into the main text.
>
> We hope that our response addresses the reviewer’s concerns and would be interested to hear if they are considering raising their score to help increase the visibility of our work during the conference. In case there are any remaining questions, we are of course happy to resolve them during the discussion period.
>
>
>
> ---
>
> Penedo et al. (2024), *"The FineWeb Datasets: Decanting the Web for the Finest Text Data at Scale"*, https://arxiv.org/abs/2406.17557
>
> Schlag et al. (2021), *"Learning associative inference using fast weight memory."*, https://arxiv.org/abs/2011.07831

---

> > ### Comment · Reviewer_TEck · 2025-08-06
> > **Thanks for your response**
> >
> > Thanks for the detailed response. This addresses some of my concerns, and I will update my score accordingly.

---

### Decision · Program_Chairs · 2025-09-17

**Decision:**

Accept (spotlight)

**Comment:**

The paper investigates a paramterizations of dense linear RNNs as fixed points of diagonal linear RNNs which enables a tunable trade-off between expressivity and efficiency. The reviewers praise the clarity of the presentation and that it is easy to follow. They also point out that  fixed-point RNNs are an elegant way to interpolate between diagonal and dense RNNs, enabling adaptive computation. During the discussion period, the authors clarified several important questions. Overall, the feedback for this paper is very positive and I recommend acceptance.